# Dynamic Amplification of Railway Bridges under Varying Wagon Pass Frequencies

Aminur K. Rahman *, Boulent Imam and Donya Hajializadeh

School of Sustainability, Civil and Environmental Engineering, Faculty of Engineering and Physical Sciences, University of Surrey, Guildford GU2 7XH, UK; b.imam@surrey.ac.uk (B.I.); d.hajializadeh@surrey.ac.uk (D.H.)
* Correspondence: a.rahman@surrey.ac.uk

**Abstract:** Train configurations give rise to a primary wagon pass forcing frequency and their multiples. When any one of these frequencies coincides with the natural frequency of vibration of the bridge, a resonant response can occur. This condition can amplify the dynamic response of the bridge, leading to increased levels of displacement, stresses and acceleration. Increased stress levels on critical bridge structural elements increases the rate at which fatigue damage accumulates. Increased bridge acceleration levels can affect passenger comfort, noise levels, and can also compromise train safety. For older bridges the effects of fatigue, and being able to predict the remaining life, has become a primary concern for bridge engineers. Better understanding of the sensitivity of fatigue damage to the characteristics of the passing train will lead to more accurate remaining life predictions and can also help to identify optimal train speeds for a given train–bridge configuration. In this paper, a mathematical model which enables the dynamic response of railway bridges to be assessed for different train configurations is presented. The model is based on the well established closed from solution of the Euler–Bernoulli Beam (EBB) model, for a series of moving loads, using the inverse Laplace–Carson transform. In this work the methodology is adapted to allow different train configurations to be easily implemented into the formulation in a generalised form. A generalised equation, which captures the primary wagon pass frequency for any train configuration, is developed and verified by presenting the results of the bridge response in the frequency domain. The model, and the accuracy of the equation for predicting the primary wagon pass frequency, is verified using independently obtained measured field train–bridge response data. The main emphasis of this work is to enable the practicing engineer, railway operators and bridge asset owners, to easily and efficiently make an initial assessment of dynamic amplification, and the optimal train speeds, for a given bridge and train configuration. This is visually presented in this work using a Campbell diagram, which shows dynamic amplification and compares this with those calculated based on the design code, across a range of train speeds. The diagram is able to identify train speeds at which a resonance response can occur, and the wagon pass frequency, or its multiples, which are causing the increased dynamic amplification. The model is implemented in Matlab and demonstrated by analysing a range of short- to medium-single span simply supported plate girder railway bridges, typically found on the UK railway network, using the standard BS-5400 train configurations. The model does not consider the effects of the train mass and suspension system as this would require a non-closed form numerical solution of the problem which is not practical for the purposes of an initial assessment of the train–bridge interaction problem.

**Keywords:** Euler–Bernoulli beam; railway bridge; wagon pass frequencies; dynamic amplification; resonance; frequency response



## 1. Introduction

In many countries, high-speed railway networks are becoming a key long-term strategy for economic growth. This is driven in part by the economic growth in urbanisation, which have resulted in continuous rise in passenger numbers, as well as the modal shift

that is being pushed towards achieving net-zero targets. To sustain this growing demand, governments are looking to increase railway capacity, firstly by upgrading existing railway networks, as well as introducing new high-speed lines. In the UK, by the year 2040, it is anticipated that an extra 101 miles of high-speed lines will be built, catering for trains which are capable of traveling at 185 mph [1]. Under the proposals there will also be a further 127 miles of fast lines, allowing trains to travel at speeds between 125 and 155 mph [2]. These upgrades are needed to maintain and increase the economic capacity of the UK, enabling it to compete on productivity with other European countries, who have established high-speed rail networks. The introduction of new high-speed rail networks and the increase in capacity on existing lines, both passenger and freight trains, may raise structural concerns for existing bridges with respect to their remaining fatigue life. As a result, the study of railway bridge dynamic effects and its impact on fatigue life is becoming an area of increased interest, particularly for ageing bridges.

There are approximately 16,000 metallic railway bridges in the UK, many of which are already exceeding their design life [3]. Many of the early railway bridges were constructed at the end of the 19th and beginning of the 20th century and had operational train speeds which did not exceed 50 km/h, whilst modern trains can now operate up to significantly higher speeds [4]. With the increased trains speeds, volume of traffic and increased axle loads, bridges may have experienced higher rate of fatigue damage accumulation on structural members over time. This highlights the importance of bridge asset owners establishing accurate fatigue life predictions to assist with the prioritisation of maintenance, and where necessary, the replacement of bridges at the end of their effective life.

## 1.1. Railway Bridge Dynamic Amplification Factor

The dynamic response of a railway bridge is governed by a combination of parameters which include the bridge's structural characteristics, train and wagon configurations, axle weights and spacing and train speeds [5]. When a train traverses a bridge, it applies a set of steady moving loads to the bridge, as well as dynamically exciting the response of the structure. As a train set is generally composed of similar repeating vehicles, all with the same axle configuration, this subjects the bridge to periodic loading. For freight trains, however, this is not always the case and the standard train mixes provided in BS-5400 [6] provide a mixed freight train for assessment purposes. The vibration of the bridge is affected by two primary mechanisms, the quasi-static loading and dynamic excitation [7]. The quasi-static loading, due to the weight of the train, is independent of train speed, whilst the dynamic excitation arises due to changes in stiffness and the irregularities at the wheel/rail interface [8]. Whilst the quasi-static load is independent of speed, the frequencies associated with the steady series of moving axle loads are a function of the train speed. It is the periodic nature of the axle loads on railway bridges which adversely affects the bridge, as it increases the rate at which fatigue damage accumulates on structural members and connections, due the cyclic nature of the stress. The severity of the effect is dependent on the train speed, particularly at conditions of resonance, where dynamic amplification of the bridge response can significantly increase the stress range on a structural element, and thereby further accelerate fatigue damage accumulation. Resonance occurs when the frequency of the periodic axle loads is in close proximity, or coincides, with the bridge's fundamental natural frequency of vibration, which in most cases is the vertical bending mode of vibration. The fundamental periodic frequency of the moving load, which is referred to as the primary wagon pass frequency, is a function of the train configuration and speed. Resonance can also be initiated by multiples of the wagon pass frequency, as is shown in this work.

At conditions of resonance, even small increases in stress levels can have a significant effect on fatigue damage since the latter is inversely proportional exponentially to the magnitude of the stress ranges. Better understanding of the wagon passing frequencies and their resulting dynamic amplifications, for various types of train traffic running on a particular route, is, therefore, important for accounting their effects on long-term fatigue

damage accumulation. Conditions of resonance for a particular train–bridge configuration could be managed operationally by knowing the critical speeds at which resonance occurs and then optimal train speeds can be specified accordingly.

Modern bridges, with their slender designs, typically have lower damping, partly due to the current design and construction methods. Therefore, these types of bridges can be more susceptible to dynamic amplification. Dynamic assessment of such bridges is performed to ensure that the bridge can safely and economically function for the required design life. In current design assessments, the acceleration and deflections of the deck are calculated and compared with allowable deck accelerations and deflection limits, as defined by the relevant design standards. The effects of the dynamic behaviour of railway bridges, particularly in estimating stresses for fatigue assessment, are accounted for in the design codes by using the Dynamic Amplification Factor (DAF) of the static response. The DAF is simply the ratio between dynamic and static responses. The DAF is also referred to by a number of other terms such as impact factor and impact coefficient. The estimation of the DAF depends on a number of parameters which govern the interaction effects between the train and bridge.

The method of calculating the DAF's in the design or assessment codes has been established through empirical means from the results of field tests and analytical studies. Hunley [9] and Looney [10] performed some of the very early analytical studies and field tests on railway bridges to bridge impact allowance due to the interaction of trains. These impact allowance factors were subsequently introduced into the early bridge design codes. These early studies related, empirically, the impact factor to a single parameter of the bridge, either the span of the bridge or its resonant frequency. This provided the basis in the design codes for calculating a global dynamic amplification factor.

These early studies paved the way for other future investigations on the subject, both in the US, by a sub-committee of the American Engineering Association [11] and in the UK, by the Bridge Stress Committee [12]. These committees were particularly concerned with the increased weight of trains and what proportion of the railway bridge stock at the time was inadequate to accommodate the increased weight. Therefore, the committees' role was to provide a better understanding of the effects of hammer blows and oscillations in both locomotive springs and bridges. It was also at this time that extensive theoretical investigations were also initiated at Cambridge University (UK), under the direction of Prof. C. E. Inglis. The committee tested a number of plate girder and truss bridges of various spans. The work of these committees, using both field test data and theoretical studies, enabled them to arrive at impact allowances included in bridge design codes.

Currently, the calculation of the DAF is provided in the various bridge design and assessment codes. In the Eurocode, EN1991-2 [13], the DAF calculations are based on the functions of train speed, bridge natural frequency and span lengths. The latter is used for new bridges and the former for existing bridges. The code also prescribes the need for performing a detailed dynamic analysis for train speeds exceeding 200 km/h and where the bridge natural frequency falls outside of the given limits, as defined by frequency limits chart in the code. In the UK, railway bridge design and assessment were performed in accordance with the requirements of BS-5400 [6]; for design of new bridges, this has now been superseded by the Eurocode. Analytical expressions used for the assessment of DAFs for existing railway bridges are provided for fatigue limit states in the Network Rail assessment code, NR/GN/CIV/025 [14]. The code provides guidance for the assessment of under-bridges and culverts and is applicable for train speeds up to 125 mph (for passenger) and 75 mph (for freight) trains. In general, the codes provide a method for calculating a dynamic amplification factor by which the stresses at critical points, using a quasi-static analysis, can be scaled to take into account dynamic effects. However, the validity of the dynamic amplifications for current rail traffic has been the subject of investigation for many years and continues to be investigated due to the complexity of train–bridge interaction effects.

## 1.2. Modelling Railway Bridge Dynamic Response

The dynamic response under moving loads requires complex modelling to represent the interaction of the bridge and train. The first investigations of railway bridge vibrations under moving loads date back to the middle of the 19th century [15]. Contributions by Willis [16] and Stokes [17] on the dynamic response of bridges resulting from a moving load were amongst some of the first researchers who paved the way for subsequent engineers to investigate this phenomenon in more detail. Prominent researchers investigating vibration and dynamics included Timoshenko [18], Jeffcott [19] and Inglis [20], who provided a general treatment on the dynamics of railway bridges.

These early studies of the moving load problem mainly focused on developing analytical and approximate solutions to simplified dynamic problems, primarily for the assessment of impact. The moving load model was then extended to replace the moving force with a moving mass. Introducing of a moving mass made the problem more difficult to solve, and to date, only numerical solutions for this purpose are available [21]. The modelling progressed on to introducing the inertia of the beam, as a distributed mass, with a massless force traversing across it. This was then further extended to include both the inertia of the load and beam. The second model, where the force was replaced with a mass known as the Willis–Stokes problem, was first formulated by Willis in 1849 [16] and subsequently solved analytically by Stokes [17]. By neglecting the beam's mass, Willis presented the fourth order partial differential equation, which represented the moving mass problem [22]. It was these early works that paved the way for more complex models of the train–bridge interaction and these eventually provided the design provisions that are available in the bridge design codes. The next step in the evolution of the moving load and mass problem was the introduction of the vehicle itself, supported by a spring damper system.

Researchers such as Fryba [23] have made significant contributions in the subject area of vibration of solids under moving loads. Other researchers, most notably, Yang et al. [24], provided closed form analytical formulations of railway bridge dynamic problems, which further expanded the work. Yang principally focused on high-speed railway bridges and provided a broad and systematic assessment of the problem of moving loads coupled with the dynamic interaction with the bridge. Their works, with the availability of modern computing power, paved the way for more advanced models. These are now widely used to investigate the dynamic train–bridge interaction effects and the various parameters that govern bridge dynamic response. As a result, numerous publications are now readily available in the literature, which have specifically looked into the aforementioned parameters as well as other specific areas such noise and passenger comfort; see for example [25].

## 1.3. Studies on Railway Bridge Dynamic Amplification

Complex train models, including coupled trains and their interaction with the dynamic response of the bridge, have been proposed in the literature. These have provided a much more detailed insight into the train–bridge dynamic interaction problem, such as the research of Karoumi et al. [26], Kwark et al. [27] and Wiberg [28]. More recently, Hamidi and Danshjoo [29] presented a parametric study on four bridges to assess dynamic amplification for different speeds, axle distance, number of axles and span lengths. The assessment was performed using a moving concentrated loads model, based on the equations provided by Yang et al. [24]. The assessment showed that, for speeds up to 300 km/h, impact factors were affected by train speed and the ratio of axle distance to span length. The dynamic response was shown to be much more significant than the static case. Impact factors were also shown to be different in three velocity ranges: <180 km/h, 180–300 km/h and >300 km/h. For speeds above 300 km/h, impact factors were shown to be more pronounced. The study also showed that, for increasing values of bridge span, impact factors under some conditions can be greater than those for shorter span bridges.

Imam and Yahya [30] estimated DAFs for critical structural elements of a railway truss bridge using a 3D FE model under the passage of a typical freight train represented

by moving forces. The effect of the DAF on the accumulation of fatigue damage was quantified by comparing the damage estimates obtained in the assessment with those obtained by the use of bridge design/assessment codes. Fatigue damage ratios were estimated to provide insight into the differences in fatigue damage estimates obtained using dynamic stress histories with those obtained by modifying the static stress histories through the DAFs. Their results showed that different bridge members experienced different magnitudes of DAF and, for speeds up to 50 km/h, the DAFs were found to agree well with the design/assessment codes. Above this speed, however, DAFs were found to be overestimated. Bisadi et al. [31] presented a parametric study of the effects of speed, natural frequency, span, axle spacing and train suspension on the dynamic response of a steel railway viaduct. The trains were simulated as a sequence of rigid sprung masses. Using an eigenvalue analysis, the work showed that DAF values increased with speed but were more prominent for shorter spans, larger axle distances and lower bridge natural frequencies. Mensinger et al. [32] presented a study of a historic railway truss bridge in Germany, which had exceeded 120 years of age but was still in service. Their assessment compared measured field stresses with those obtained theoretically on critical elements of the bridge. The authors showed that by ignoring the bridge and train dynamic effects, the DAFs obtained using bridge design codes may underestimate the fatigue life for an ageing bridge. For new-built short-span bridges, the authors show that the DAFs obtained using the design codes are on the safe side, but they highlighted that caution is required when assessing existing historic bridges.

Numerous other publications have shown that, for bridges subjected to periodic loads, the associated wagon pass frequencies are one of the primary dynamic parameters which can govern the dynamic response of a bridge and increase dynamic amplification. Train wagon pass frequencies give rise to two types of frequencies, "dominant frequencies" and "driving frequencies" [5]. The dominant frequencies are associated with the periodic loading from the axles and the driving frequency is associated with the duration of a train wagon/carriage passing over the bridge. It was shown that the severity of resonance at a particular speed could be assessed by a parameter, termed the Z factor, which is a function of the bridge-to-carriage length ratio [5,33]. By considering the moving mass, an additional term was also introduced, i.e., an effective natural frequency of the bridge. This is a reduced first mode bending natural frequency of the bridge because of the moving mass. The investigation of wagon pass frequencies, with particular reference to the variation in the natural frequency of the bridge under laden trains, has been the subject of investigation by a number of other researchers in the past [34–38]. More recently [7], the study of the properties of train load frequencies from measured track vibration spectra has been carried out. This study showed that the most prominent frequencies occur at the integer multiples of the wagon pass frequency. For trains that consist of a single wagon type, the train load spectrum depends on the number of vehicles and their geometry. Other researchers, who have carried similar studies, include Auersch [39], Ju et al., [40] and Gatti et al., [41], who have studied the frequency spectra of track vibration to identify the dominant train loading and wagon passing frequencies.

The study of dynamic amplification on railway bridges has received considerable interest in recent times. This has been primarily driven by the need for assessing existing ageing railway bridges due to the increase in rail traffic, axle loads and increased train speeds, which are not explicitly accounted for in the design codes. The dynamic response of high-speed railway bridges has been of high interest, particularly throughout Europe [42]. Furthermore, with advances in mathematical modelling techniques and the power of modern computing, it has become easier to assess the many parameters of the train–bridge interaction which could affect bridge dynamic amplification. Various studies have shown that dynamic amplification can result in an overly conservative estimate of fatigue life or an underestimate, depending on the actual dynamic response of the bridge [30].

### 1.4. Train–Bridge Interaction (TBI) Models

The dynamic modelling of train–bridge interaction effects has evolved considerably in recent years through the use of more complex train models utilising both finite element modelling (FEM) and numerical methods. These models capture the dynamics of the train and rails, as well as the bridge [43–49]. The train models in these studies included the inertia of the carriage, bogies, wheels and the primary and secondary suspension systems. In some cases, track irregularities were also included in the model [43,44]. In these studies, by using the Lagrange method, the authors derived a model for studying the dynamic interaction between a 10-DOF high-speed train model on a railway bridge. The bridge was represented as a simply supported beam based on the Euler–Bernoulli beam formulation. The model was shown to be more efficient than other train–bridge interaction (TBI) models based on FEM. The 10-DOF lumped parameter system represented a half train model moving at a constant speed along the beam. The 10 two-degree second order equations were reduced to a system of twenty first degree equations and solved using a fourth order Runge–Kutta method.

The effect of vehicle mass was investigated in [44] where it was shown that an increase in train mass increased the displacement response of both the vehicle and bridge. The study also showed that there exists a phase difference between the forcing frequency of the train and the response of the train and bridge. These models were investigated previously by other researchers [50–52]. The same methodology was extended in [45] to include the study of high-speed trains based on the 10-DOF system model. In this case, two different numerical methods were employed with the bridge represented as a beam using both the Euler–Bernoulli theory (EBT) and Timoshenko Beam theory (TBT). The differential equations were solved in the time domain using a fourth order Runge–Kutta (RK) method. The motion equations were also converted into the finite element format and solved using the Newmark-beta method. A comparison of the results for both solution methods showed good agreement with the FEM solution approaching close to the RK solution when the number of elements were increased.

In [48], the response of a railway bridge to moving loads is presented. The authors provided a non-linear vehicle-interaction model based on a two systems approach and compared the response of the bridge using both moving sprung mass and moving load models. The solution to the problem was obtained in the FE numerical domain in two methods. In the first method, the system was treated as one whole system where the stiffness matrix is updated dynamically and the solutions are obtained non-iteratively at every time step. The second method considers the contact forces between two decoupled systems using constraint equations. This method is iterative and used Newmark's-beta time integration method. The bridge was represented as an Euler–Bernoulli beam with the tracks modelled as a single continuous non-viscous elastic layer beneath the rails, accounting also for rail irregularities. The train system was treated as a separate moving sprung-mass system using a full model of car body and suspensions, reduced to a lumped mass and one suspension layer model. The key finding from this work was that the moving load model was shown to overestimate the response of the bridge at resonance conditions and the frequency was also shown to be slightly higher. The stiffness of the train suspension was found to have minimal effect on the response of the bridge.

All of above studies have primarily focused on the vehicle dynamics and the corresponding interaction forces on the system. However, these numerical methods do not provide a closed form solution and, because of the large numbers of degrees of freedom involved in the formulation of the problem, they may not be computationally efficient, nor easy to implement in practice. As discussed, numerical models used to investigate train–bridge interaction dynamics included the finite element methods, the fourth order Runge–Kutta solution and the iterative Newmark's time integration method. The finite element method is by far the most widely used in the literature. These modelling approaches require more time for the subsequent running of the loading cases and investigation of the many different parameters that constitute a train–bridge interaction model. Although

Runge–Kutta methods are generally stable and have good accuracy, they can require considerable solution times, particularly for problems with a high number of degrees of freedom, as is the case with many of the vehicle–bridge interaction models. Iterative numerical methods, such as the Newmark's time integration method can also be subject to numerical instability and errors, which are oscillatory in nature, increasing as the time step increases [53].

### 1.5. The Need for Further Work and Simplified Models

Regardless of the theoretical advances in the various ways of modelling train–bridge interaction, such complexity may prohibit a quick initial fatigue assessment of railway bridges. Such advanced models are more appropriate when considering noise and vibration-related wear but whether they are suited for the assessment of fatigue damage has not been addressed in the literature. Fatigue damage may be particularly sensitive to the dynamic amplification of the stress response of bridge members and connections. This is an area that further investigation is needed, particularly for the large ageing and historic railway bridge stock in Europe and other parts of the world. In practice, having the means of assessing the influence of railway bridge dynamic effects on fatigue life relatively quickly, using simple practical models, would be invaluable to bridge owners prior to engaging with any complex time-consuming modelling. This need has been noted by some researchers who have re-visited the moving load problem [54,55], demonstrating that such models are still able to provide the bridge engineer with vital information. For example, estimation of dynamic amplification under different conditions and a subsequent initial fatigue assessment can help towards identifying critical parameters for further detailed investigation, such as the most fatigue damaging train configurations or locations on the bridge. The model itself can also provide the inputs for developing a more detailed localised FE model of critical bridge details/locations.

Evidence suggests that simple models are valuable, as long as they are able to provide and identify key information on any potential dynamic issues such as resonance effects. Resonance effects, for higher train–bridge mass ratios, can also be assessed with modifications to the moving force problem formulation. Simple moving force models offer a quick and efficient closed form solution to the bridge-train dynamic problem. They can also offer other useful information for the engineer. For example, using simple numerical calculations, other desired characteristics of the problem such as dynamic amplification, fatigue damage and amplitude-velocity relationships can be obtained without any additional significant computational effort required for the solution of the problem.

By revisiting the moving load problem, a comprehensive Matlab-based model is presented in this paper, by which any bridge and train configuration can easily be assessed quickly and efficiently to predict dynamic amplification factors and identify train critical speeds. The train–bridge interaction model presented in this paper is based on the Euler–Bernoulli Beam theory, which describes the behaviour of a simply supported beam under the action of a moving load, as presented by Fryba [23] and Yang [24]. The model is adapted to allow for different train configurations to be easily implemented, as a series of moving loads in the formulation in a standard generalised form. The fourth order partial differential equations are solved using the inverse Laplace–Carson integral transform, which provides a closed form solution to the problem. A generalised equation, which enables the primary wagon pass frequency for any train configuration, is presented and verified by presenting the results of the bridge response in the frequency domain. The model and the accuracy of the equation for predicting the primary wagon pass frequency is verified using independently obtained measured field train–bridge response data. The calibration of the model has also been carried out using FE analysis.

As the model provides a closed from solution it is extremely efficient and allows a database of different bridges and train configurations to be added. The model can provide vital information and has the scope to be extended to provide other useful information on

the train–bridge dynamic interaction, particularly in generating loading or stress histograms for more detailed fatigue analysis.

The application of the developed model is demonstrated by performing frequency response analyses on typical metallic plate girder bridges under the actions of different train configurations using the BS-5400 [6] medium train mixes. The paper also investigates, explicitly, the variation in DAFs on the bridge with respect to different wagon passing frequencies and compares the findings with the DAFs given in bridge codes.

## 2. Bridge Moving Load Models

The dynamic moving load model used in this paper is based on the Euler–Bernoulli Beam (EBB) theory, as formulated by Fryba [23]. The model is implemented within Matlab and the dynamic response results are compared with a quasi-static moving load model using standard beam theory through applying the principle of superposition to establish the Dynamic Amplification Factors (DAFs) for different train configurations and speeds.

### 2.1. Euler–Bernoulli Beam (EBB) Model

The representation of the bridge response under a series of moving loads is depicted in Figure 1. The problem can be solved using the classical Euler–Bernoulli Beam theory, represented by Equation (1), for a single moving load, $P$, traversing a beam [56].

$$\sum_{i=1}^{\infty} \left( EI \frac{\partial^4 \phi_i(x)}{\partial x^4} q_i(t) + \mu \frac{\partial^2 q_i(t)}{\partial t^2} \phi_i(x) + c \frac{\partial q_i(t)}{\partial t} \phi_i(x) \right) = \delta(x - vt)P \tag{1}$$

| | |
|---|---|
| $EI$ | Flexural rigidity of the beam with a constant moment of inertia, |
| $\phi_i(x)$ | Linear combination of normal modes, |
| $q_i(t)$ | Generalised coordinate of the $n^{th}$ mode, |
| $x$ | Length coordinate from origin—right hand of the beam, |
| $t$ | Elapsed time from the instant at which the moving concentrated load P enters the beam, |
| $\mu$ | Mass per unit length of the beam, |
| $c$ | Equivalent coefficient of viscous damping of the beam, |
| $\delta$ | Dirac delta function which describes moving concentrated load, |
| $v$ | Load travelling speed, |
| $P$ | Moving concentrated load. |

This equation describes the motion of a beam with a flexural rigidity $EI$ and a uniformly distributed mass; $\mu$, along which the force, $P$, moves at constant speed, $v$. The parameters on the right-hand side of Equation (1) represent the motion of the constant force, which is described by the Dirac function $\delta(x)$ [57]. The function of the vertical deflection of the beam, $y(x, t)$, can be expressed as a product of two functions, the mode shape function (Eigen-function), $\phi_i(x)$, and the function of the generalised coordinates, $q_i(t)$.

Equation (1) forms the basis of the train–bridge interaction model for a series of moving loads [23,24]. For a series of moving loads, the equation is extended to the form given by Equation (2).

$$EI \frac{\partial^4 y(x,t)}{\partial x^4} + \mu \frac{\partial^2 y(x,t)}{\partial t^2} + 2\,\mu\,\omega_d \frac{\partial y(x,t)}{\partial t} = \sum_{n=1}^{N} \varepsilon_n(t)\,\delta\,(x - x_n)\,F_n \tag{2}$$

| | |
|---|---|
| $y(x,t)$ | Vertical deflection of the bridge at position x and time t, |
| $\omega_d$ | Circular damped frequency of the bridge, |
| $\varepsilon_n(t)$ | Describes the Heaviside unit step function for the arrival (turning on) and departure (turning off) of the $n^{th}$ axle force, $F_n$, |
| $F_n$ | Constant magnitude concentrated axle force, |
| $X_n$ | Position of the nth axle force, $F_n$, from the first axle, |
| $v$ | Train constant speed, |
| $x_n = vt - X_n$ | Position of the nth axle force, $F_n$, from the bridge origin. |

Equation (2) is solved using the fundamental relations of the Fourier sine integral transformation which presents the problem in the frequency domain. The method of Laplace–Carson Integral transformation is then applied to present the problem in the complex domain. The inversion of the Laplace–Carson transforms then presents the problem in the real space and the Fourier transform is then used to reduce the equation to the time domain. This method enables an analytical closed form solution of Equation (2). This method has been applied by Fryba [23] to determine the closed form solution, enabling the calculation of the vertical deflections of the bridge at any specific location $x$ along the bridge as a function of time as given by Equation (3). The equation expresses the forced vibration of the bridge due to the moving loads and the free transient damped vibrations after the train has left the bridge. The acceleration response of the bridge can be obtained by double differentiation of Equation (3).

$$y(x,t) = \sum_{j=1}^{\infty} \sum_{n=1}^{N} y_0 F_n j\omega\omega_1^2 \left[ f(t-t_n)H(t-t_n) - (-1)^j f(t-T_n)H(t-T_n) \right] \sin\frac{j\pi x}{L} \quad (3)$$

| | |
|---|---|
| $y_0$ | Unit load deflection, |
| $L$ | Bridge span, |
| $j$ | $j_{th}$ Modal frequency (j = 1 for first vertical bending mode), |
| $\omega$ | Forcing frequency, |
| $\omega_1$ | Circular natural frequency of vibration of the bridge (1st vertical bending mode). |

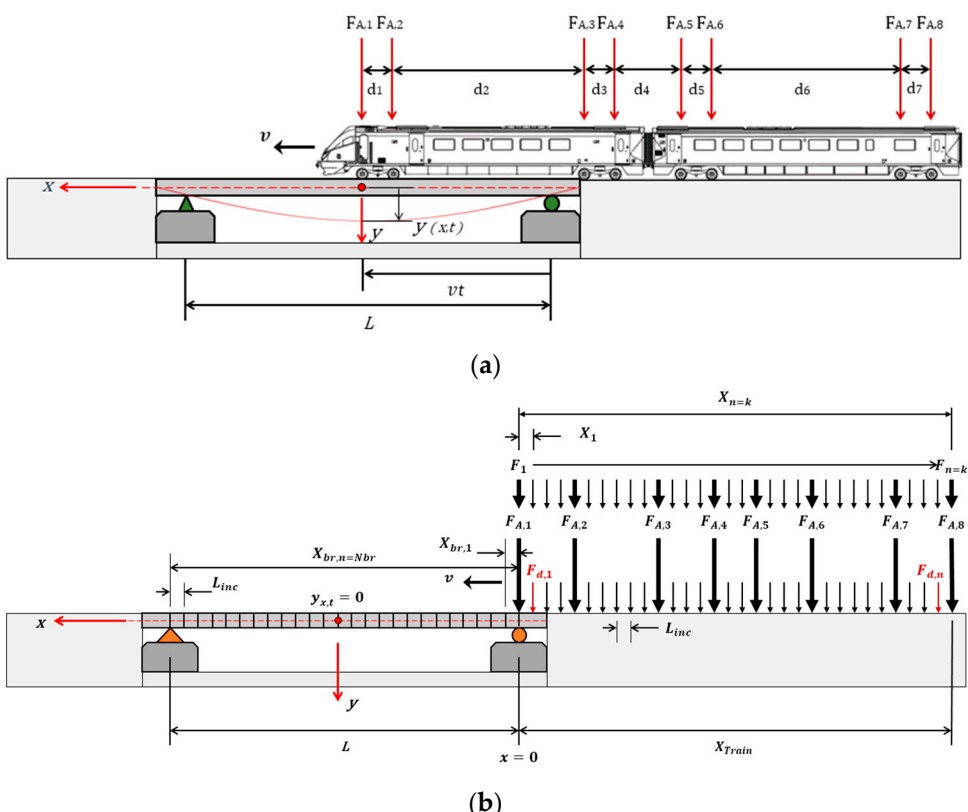

**(a)**

**(b)**

**Figure 1.** (**a**) Bridge under moving load, (**b**) idealisation for mathematical modelling.

The time when the $n^{th}$ axle force, $F_n$, enters the bridge is given by Equation (4), and the time when it leaves, by Equation (5).

$$t_n = \frac{X_n}{v} \quad (4)$$

$$T_n = \frac{L + X_n}{v} \tag{5}$$

For the closed form solution, Equation (6) describes the Heaviside unit step function, $H(t)$, for the arrival (turning on) and departure (turning off) of the $n^{th}$ axle force, $F_n$, and their time shifts $t - t_n$ and $t - T_n$, respectively.

$$\varepsilon_n(t) = H(t - t_n) - H(t - T_n), \quad H(t) = \left\{ \begin{array}{ll} 0, & t < 0 \\ 1, & t \geq 0 \end{array} \right. \tag{6}$$

The parameter $f$ in Equation (3) is a function of the Inverse Laplace–Carson transformation, as given by Equation (7). The first term expresses the response of the bridge due to the moving loads, and the second term, is the transient response.

$$f(t) = \frac{1}{\omega_j' D} \left[ \frac{\omega_j'}{j\,\omega} \sin(j\omega t + \theta) + e^{-\omega_d\, t} \sin\left(\omega_j'\, t + \varphi\right) \right] \tag{7}$$

Further details on the parameters and derivation of the equations can be found in Fryba [23]. The model equations were implemented and solved within Matlab (Version R2023). The quasi-static and dynamic model Matlab algorithms, including the inputs and outputs, are described by the flow chart shown in Figure 2.

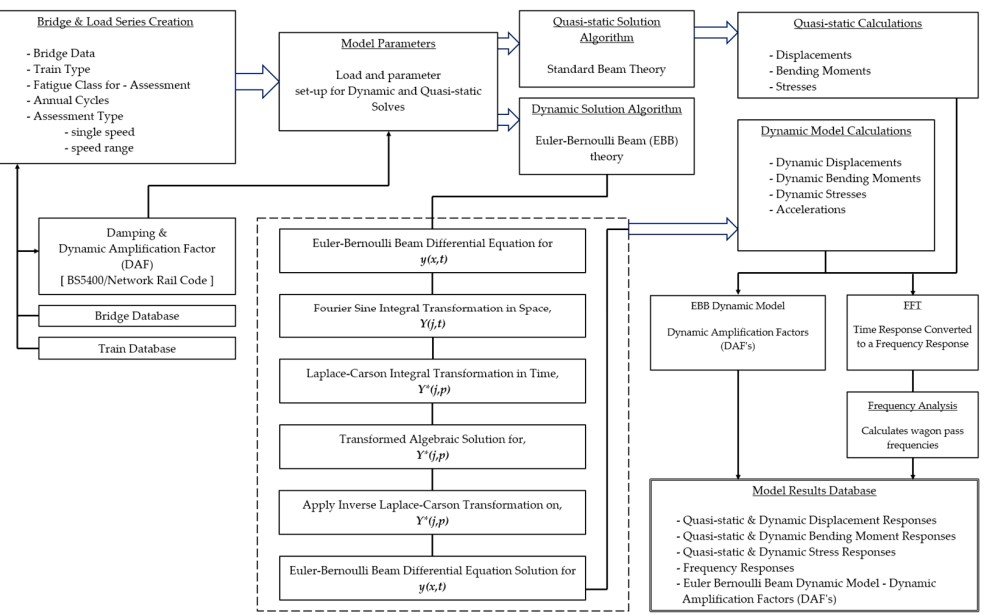

**Figure 2.** EBB dynamic and quasi-static analysis model flow chart.

### 2.2. Case Study Plate Girder Bridges

The bridges selected for analysis in this study represent a high proportion of the existing bridge stock in the UK [58]. These range between 8 m and 21 m span, as shown in Table 1, which is representative of the typical span ranges that can be found on the UK rail network. The span consideration is important so that the effect of the train axle configuration is adequately captured in the bridge response. Another important consideration is the range of bridge mass and stiffness, and ensuring that the selected bridges are also representative of typical bridge types in terms of their mass and stiffness. The bridge natural frequencies also fall within the frequency range limits defined for unloaded bridges in the BS-5400 bridge assessment code [6,14]. The selected bridges fall into the category of Low Frequency, Light Mass and Medium Damping [58].

**Table 1.** Case study plate girder bridges used in analysis [58].

| Bridge No. | Bridge Type | Bridge Span, *L* [m] | Bridge Mass, *M* [kg] | Vertical Bending Frequency, $f_n$ [Hz] | Second Moment of Area, *I* [m⁴] |
|:---:|:---:|:---:|:---:|:---:|:---:|
| 1 | Half-through | 8.84 | 42,400 | 10.5 | 0.0062 |
| 2 | Half-through | 18.1 | 133,200 | 5.3 | 0.0428 |
| 3 | Western Box and Half-through deck | 9.3 | 115,000 | 14 | 0.0350 |
| 4 | Western Box and Half-through deck | 21.33 | 400,600 | 6.8 | 0.3468 |
| 5 | Half-through | 8.1 | 55,527 | 12.1 | 0.0083 |
| 6 * | Half-through | 21.26 | 207,832 | 5.5 | 0.1166 |

* Bridge data taken from Gu [59].

The case study bridges are of the plate half-through and box girder type construction and cover a range, in terms of their span and structural parameters, that could potentially result in significant dynamic responses. Since the mathematical model captures a global analysis of the bridge response in terms of vertical bending, only information required to simulate a bridge as an equivalent simply supported beam is required. Therefore, the data in Table 1 provides information for this purpose only. Information on the track, ballast and train mass and suspension parameters have not been included as these are not accounted for in the mathematical model. Although this can be considered as a limitation, the model is still deemed suitable for a quick, initial assessment of the parameters considered in this study. On the other hand, a useful parameter which is specified in bridge design/assessment codes, allowable bridge acceleration limits, can be obtained as global response using the proposed model.

The selected bridge types have shallow construction depths and comprise either two or three main longitudinal girders and transverse steel girders. Bridges 1 and 2 have two main longitudinal girders and the transverse girders are encased in concrete fill. For Bridge 3, the inner main girder is shared by the two adjacent half-through decks. In Bridges 4 and 5 the steel cross girders support a steel floor plate. Typical cross-sections of half-through and box girder type bridges are shown in Figure 3. The lateral stability of the main-girder top flanges, which are in compression, is achieved by U-frame action. For the purposes of these analyses, only the bridge parameters required to calculate the first vertical bending mode of the bridge are of interest, and these are provided in Table 1.

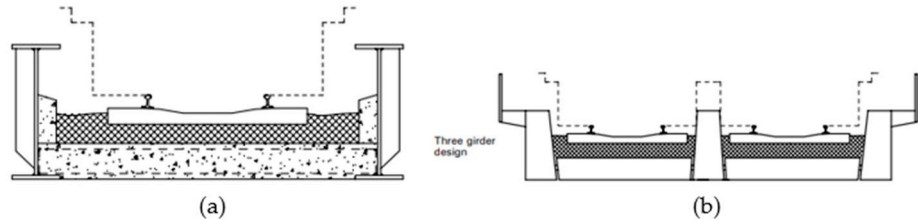

**Figure 3.** (**a**) Half-through deck and (**b**) box girder type bridges [60].

### 2.3. Dynamic Amplification Factor (DAF)

In the literature there are different definitions used to define the DAF resulting from the interaction of moving vehicles and the bridge structure. Chan et al. [61] describe the DAF in terms of the static equivalent of the dynamic and vibratory effects. Ichikawa et al. [62] describe the DAF using the ratio of the dynamic deflection obtained from the moving mass to that of the moving force problem. Brady et al. [63] describe the DAF in terms of the ratio between the maximum dynamic and static responses. In the comparison of the DAF between continuous and simply supported bridges, Mohamed et al. [64] define the

DAF as the ratio of the maximum total load effect (static + dynamic) to the maximum static load effect for a given section of the bridge. There are also other terms used in the literature by which a bridge's dynamic amplification is characterised such as 'dynamic increment' [65], 'impact factor' or 'dynamic load allowance' [66], 'dynamic increment factor' [67] or 'dynamic load allowance' [61].

In this study, the DAF is defined as the ratio between the bridge's maximum dynamic to static responses for different load effects resulting from the passing train. This is defined in terms of deflection according to Equation (8), where the peak values of $y_{(x,t),dynamic}$ and $y_{(x,t),static}$ of the dynamic and static displacement responses are used.

$$DAF = \frac{y_{(x,t),dynamic}}{y_{(x,t),\,static}} \tag{8}$$

The dynamic deflection response is obtained from the solution of the Euler–Bernoulli beam model. The static deflection response is based on a quasi-static analysis, based on the standard beam theory and by applying the principle of superposition to obtain the maximum deflection for the passing train. The results are compared with the DAF calculated using the UK railway bridge assessment code [14]. The code is applicable for train speeds up to 125 mph (passenger) and 75 mph (freight). The calculation method given in [14] provides an additional distinction between longitudinal and transverse members. In the relevant Eurocode [13], different loading models are provided for the design of railway bridges. The expressions for calculating the impact factors are based on the train speed, bridge natural frequency (for existing bridges) and span length (for new bridges). For train speeds exceeding 200 km/h, the code also prescribes a more detailed dynamic analysis.

In [14], which is used in this work for fatigue damage calculations, the dynamic increment $\varphi$, for the bending of a longitudinal member, is given by the same equations included in Eurocode 1, except for the calculation of the parameter $k$ for the basic dynamic increment, $\phi'$ and the increment for track irregularity, $\phi''$. The procedure for calculating the DAF is as follows:

$$k = \frac{v}{4.47 L_\Phi \eta_o} \tag{9}$$

$$\phi' = \frac{k}{1 - k + k^4} \tag{10}$$

$$\phi'' = \alpha \left[ 56 e^{-\left(\frac{L_\Phi}{10}\right)^2} + 50 \left(\frac{L\eta_o}{80} - 1\right) e^{-\left(\frac{L_\Phi}{20}\right)^2} \right] \quad but > 0 \tag{11}$$

where $\alpha = 0.002v$ but not $>0.01$ and $v$ is the train speed in mph, which is normally taken as the permissible speed for the bridge, $L_\Phi$ is the determinant length in metres, $L$ is span of the bridge member (centre-to-centre of supports) in metres and $\eta_o$ is the fundamental natural frequency of vibration in Hertz of the structural member based on the $\delta_o$ resulting from the uniformly distributed self-weight deflection, $w$.

$$\eta_o = \frac{17.75}{\sqrt{\delta_o}} \quad where \,\, \delta_o \,\, is \,\, in \,\, mm \,\, given \,\, by, \tag{12}$$

$$\delta_o = 1000 \frac{5wL^4}{384EI} \tag{13}$$

The upper and lower bounds of the natural frequency $\eta_o$ are estimated as follows,

$$\eta_o = 94.76 L^{-0.748} \,\, (upper \,\, bound) \tag{14}$$

$$\eta_o = \frac{80}{L} \quad (lower \,\, bound \,\, for \,\, 4m \leq L \leq 20m) \tag{15}$$

The DAF for longitudinal members in bending is then given by Equation (16).

$$DAF_{bending} = 1 + 0.5\left(\phi' + \frac{\phi''}{2}\right) \tag{16}$$

The quantity $0.5\left(\phi' + \frac{\phi''}{2}\right)$ is the dynamic increment, $\phi$.

The DAFs obtained from the dynamic analysis are compared with their bridge assessment code counterparts. The DAF estimates from the assessment code are shown in Figure 4, for each of the bridges in Table 1. For train speeds of 48 km/h (30 mph), 105 km/h (65 mph) and 201 km/h (125 mph), the DAFs are shown to be lower for bridges with a low natural frequency of vibration, which is typical for longer spans, and higher for bridges with shorter spans. The increase in DAF between the longer and shorter spans are 3%, 5% and 6% for the 48 km/h (30 mph), 105 km/h (65 mph), and 201 km/h (125 mph) train speeds, respectively. These train speeds are typical of the range of train speeds on the UK railway network.

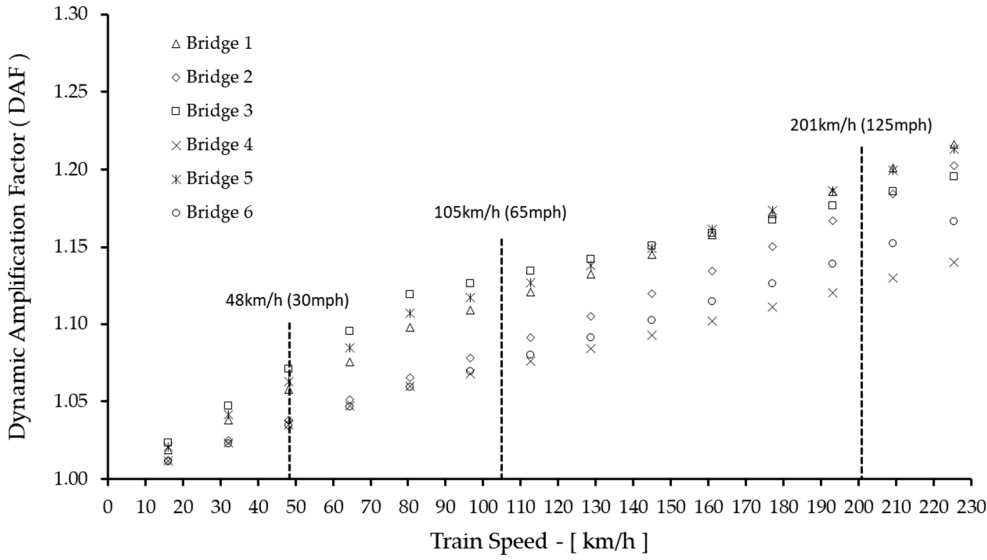

**Figure 4.** DAFs calculated according to the UK railway bridge assessment code [14] for the case study bridges.

In a dynamic analysis, the limit defined for the permissible vertical deck accelerations is often the design driver for bridges. For bridges with a ballasted track the maximum allowable bridge deck acceleration is 3.5 m/s$^2$ whereas, for non-ballasted tracks, it is equal to 5 m/s$^2$ [13]. High-bridge deck accelerations in ballasted tracks can have a destabilising effect leading to rapid track deterioration and stability. This may necessitate maintenance interventions which, if neglected, could lead to safety concerns. There are also other criteria which are affected by dynamic amplification of loads, such as bridge end rotations as well as passenger comfort. The latter is particularly important for high-speed trains where a significant amount of research has been carried out through both experimental studies and the use of complex TBI models.

Bridge design criteria based on meeting acceleration requirements, or when considering dynamic effects purely based on amplifying bridge response with a dynamic amplification factor (DAF), do not take into account bridge response under resonant conditions. By ignoring this effect bridge fatigue life could be overestimated, particularly if trains are exciting any of the bridge's natural frequencies. The train wagon pass frequencies and their multiples, which are of particular interest, are not addressed in the design/assessment codes. As this effect is specific to the particular bridge in question and the configuration and speed of the train, the codes do not specifically address this and it is left to the bridge

designer to make an assessment. Even if fatigue is not an issue, minor resonance effects can still have a detrimental impact on noise levels, which is particularly important for passenger comfort and noise pollution in urbanised areas. Wear and fretting are also other undesirable phenomena which are affected by a resonant response, even at minor levels. By capturing an assessment of a particular train type, this work can further provide vital information on these phenomena, including train speeds that are likely to affect them, thus helping engineers to make informed decisions.

### 2.4. BS 5400 Train Configurations

The BS-5400 standard train configurations (Trains 1–9) are shown in Table 2 [6]. The analytical model described in the previous section was used to obtain the dynamic response of the case study bridges (Table 1) for the BS-5400 medium traffic mix train configurations (which includes Trains 1, 5, 7 and 8). A medium traffic mix is representative of the railway traffic experienced by bridges, which includes both passenger and freight traffic. The assessment of the wagon pass frequencies is performed for these four trains only. The general train configuration is shown in Figure 5 and the parameters for each train type are given in Table 3.

**Table 2.** BS-5400 [6] standard train mixes (L: Locomotive; W: Wagon).

| Train No. | BS-5400 Train Type | | No. Axles & Weights [t] | | No Wagons | | Locomotive-Wagon (Train) Configuration |
|---|---|---|---|---|---|---|---|
| | | | Locomotive | Wagons | | | |
| 1 | Steel | S-T1 | 6 × 21.5 t | 6 × 18.5 t | 15 | W | L—15 × W |
| 2 | Electric Multiple Unit | EMU-T2 | 4 × 16.5 t | 4 × 10 t | 3 | W1 | L—3 × W1—L—3 × W2 |
| | | | | 4 × 16.5 t (Loc) | 1 | L | |
| | | | | 4 × 10 t | 3 | W2 | |
| 3 | Southern Regional Suburban | SRS-T3 | 2 × 13 t + 2 × 11 t | 4 × 9.5 t | 2 | W1 | L—2 × W1—W2—W3—2 × W1—W2 |
| | | | | 2 × 11 t + 2 × 13 t | 1 | W2 | |
| | | | | 2 × 13 t + 2 × 11 t | 1 | W3 | |
| | | | | 4 × 9.5 t | 2 | W1 | |
| | | | | 2 × 11 t + 2 × 13 t | 1 | W2 | |
| 4 | Southern Regional Suburban | SRS-T4 | 2×13 t + 2×11 t | 4 × 9.5 t | 2 | W1 | L—2 × W1—W2 |
| | | | | 2 × 11 t + 2 × 13 t | 1 | W2 | |
| 5 | Diesel Hauled Passenger | DHP-T5 | 6 × 20 t | 4 × 10 t | 12 | W | L—12 × W |
| 6 | Electric Hauled Passenger | EHP-T6 | 4 × 23 t | 4 × 10 t | 12 | W | L—12 × W |
| 7 | Heavy Freight | HF-T7 | 6 × 20 t | 4 × 25 t | 10 | W | L—10 × W |
| 8 | Heavy Freight | HF-T8 | 6 × 20 t | 2 × 25 t | 20 | W | L—20 × W |
| 9 | Mixed Freight | MF-T9 | 6 × 20 t | 2 × 7 t | 18 | W1 | L—2 × W1—W2—10 × W1—W3—W2—2 × W1—W2—4 × W1—W3 |
| | | | | 4 × 20 t | 3 | W2 | |
| | | | | 6 × 20 t | 2 | W3 | |

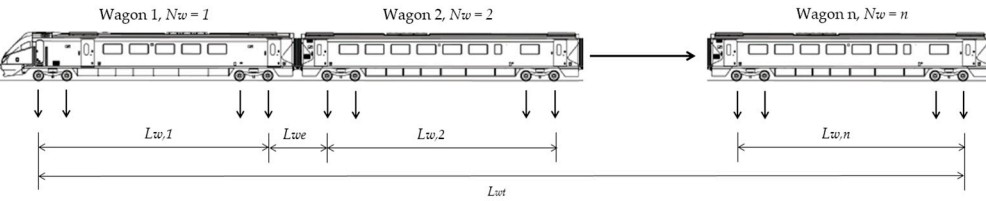

**Figure 5.** Train dimension nomenclature used in analyses.

**Table 3.** BS-5400 medium traffic mix train parameters.

| Train Parameters | | S-T1 | DHP-T5 | HF-T7 | HF-T8 |
|---|---|---|---|---|---|
| Wagon Length, $L_w$ | [m] | 11.2 | 16.7 | 15.7 | 5.5 |
| Wagon Coupling Distance, $L_{we}$ | [m] | 3.5 | 3.6 | 3.6 | 3.5 |
| Number of Wagons, $N_w$ | | 15 | 12 | 10 | 20 |

*2.5. Wagon Pass Frequency*

The wagon pass frequency is a function of the axle spacing of each train type and the number of wagons. As an entire train can comprise a different number of wagons, which can also be mixed, then this, together with their individual axle spacing and the coupling distance between the wagons, will have an influence on the wagon pass frequencies. By considering all these parameters, the wagon pass frequency, $f_{wp}$, can be calculated for any speed, $v$, using Equation (17).

$$f_{wp} = \frac{5jV}{18\left(L_w + L_{we}\left(1 - \frac{1}{N_w}\right)\right)} \tag{17}$$

$V$      Train speed in km/h,
$L_w$      Wagon spacing of two outer axles according to Figure 4,
$L_{we}$      Wagon end coupling distance,
$N_w$      Number of wagons.
$j$      Integer multiples, j = 1,2,3...n.

The term, $\left(L_w + L_{we}\left(1 - \frac{1}{N_w}\right)\right)$, can be considered as an equivalent wagon length, $L_{w,eq}$. Using Equation (17) and $L_{w,eq}$, the critical train speed at which the fundamental bending frequency, $f_n$, of the bridge will be excited, causing a resonant response which can be calculated using Equation (18).

$$V_{critical} = \frac{3.6f_n}{j}L_{w,eq} \; [km/h] \qquad f_n = natural\ frequency\ of\ bridge\ in\ Hz \tag{18}$$

**3. EBB Dynamic Model Validation**

The EBB dynamic model developed in this work provides a simple but effective means of predicting the response of railway bridges with similar profiles and structural configurations subject to moving loads. It is presented as an easy-to-implement model (Figure 2) which allows for different train configurations and bridges to be easily modelled and assessed for a number of effects such as dynamic amplification, displacements, accelerations, bending moments and stress time histories, which can be used to perform fatigue assessment.

The validation of the EBB dynamic model is presented and discussed in Appendix A. The model was verified and validated using FE analysis and measured response data from a plate-girder metallic railway bridge in operation before it was decommissioned [68]. The model was first verified to ensure that the EBB dynamic model was representative for stiffness and for the fundamental vertical bending frequency. The validation of the model was achieved based on the following:

- Verification based on standard beam theory.
- A comparison of modal response and displacement influence curves obtained from a 3D FE model of the case study bridge that was developed (Figure A1) with those obtained from the EBB dynamic model.
- Correlation of the displacement time history of the EBB dynamic model with measured response data.
- Correlation of the measured acceleration frequency response and the primary wagon pass frequency with those predicted by the model.

A number of limitations of the model include the fact that it only captures the global response of the bridge in the vertical direction and it does not consider the dynamics of the train itself or the tracks. On the other hand, the main advantage of the model is that it provides a closed form solution to a complex problem and is able to provide key information very quickly and efficiently, helping the engineer to make informed decisions on whether a further detailed assessment or investigation is needed, which is extremely useful in real life application.

The train–bridge interaction is a complex dynamic problem and depending on the parameters being investigated the current model may not be suitable. However, the model does provide scope for further enhancement to include the consideration of train mass and suspension, as well as track irregularity. How these additional parameters will affect dynamic amplification and ultimately the resulting stress histories on a bridge member/connection is of particular interest within the context of fatigue damage. It is well known that track irregularity and the dynamic response of the train can affect bridge accelerations, but it is not clear as to how these will affect stress ranges on bridge structural elements. This is an area that has not been covered adequately in the literature and therefore the approach presented in this study provides further scope to address this knowledge gap. In the following section, the model is demonstrated by predicting the effects on dynamic amplification for a range of bridges using the standard BS-5400 train types.

## 4. Results and Discussion for Case Study Bridges 1–6

The EBB dynamic model is demonstrated using six case study bridges (Table 1). The purpose of this study is to demonstrate the applicability of the model in predicting the dynamic response of the bridge, in the time and frequency domains, and dynamic amplification, using the standard BS-5400 [6] train types (Table 2). By presenting the results in the frequency domain, this helps towards identifying the main frequencies which are driving the response of the bridge and correlating them with the calculated wagon pass frequencies and their multiples. The dynamic amplification of the bridge mid-span deflection is obtained for each bridge using each of the train types and compared with the dynamic amplification as predicted by the UK railway bridge assessment code [14]. The train types used in this assessment are defined as trains S-T1, DHP-T5, HF-T7 and HF-T8 (Table 2), which represent the standard medium traffic train mixes, according to BS-5400 [6], and are typically used to perform fatigue assessment.

Train S-T1 has a locomotive followed by fifteen wagons, both having six axles. Train DHP-T5 is a passenger train, with its locomotive having six axles and the twelve wagons each having four axles. Trains HF-T7 and HF-T8 are heavy freight trains with their locomotives having six axles. There are ten wagons for HF-T7, each with four axles, and twenty wagons for HF-T8, each with two axles. The response results obtained are in the form of displacement time histories and FFT (Fast Fourier Transform) responses for each train type traversed over each of the six case study bridges. For comparison purposes, results are also shown for the quasi-static cases, which utilise the DAF calculated by the railway bridge assessment code [14] to amplify the results obtained from a static analysis using the principle of superposition. For the dynamic analysis using the EBB model and for the purposes of this study, a typical speed of 100 km/h was used for each train for the assessment of displacement response and wagon pass frequencies. To calculate dynamic amplification across a train speed range, each train was run up to a speed of 300 km/h in increments of 1 km/h. Although this upper speed magnitude exceeds the typical assessment speeds for the trains given in the assessment code [14], the purpose here is to demonstrate the effectiveness of the EBB dynamic model to predict dynamic amplification under resonant conditions. All displacement and FFT response plots are given in Appendix B. In the FFT plots (Figures A14–A17 in Appendix B), the vertical lines represent the wagon pass frequencies at integer multiples according to Equation (17).

### 4.1. Bridge Dynamic Response at 100 km/h—Train S-T1

By comparing the quasi-static analysis displacement responses for each train type with the EBB dynamic model results, dynamic amplification and resonance conditions can be assessed. For train S-T1, the displacement responses in Appendix B Figure A14a,c,e,g,i,k do not show any resonance effects. The results, however, do show that the displacements are marginally higher under dynamic response conditions. The increase in the response under dynamic conditions are not driven by the primary wagon pass frequency, when $j = 1$ in Equation (9), but by the higher integer multiples of $f_{wp}$, as shown in the frequency response plots of Figure A14b,d,f,h,j,l.

In Figure A14b the first two peaks, when $j = 1$ and $j = 2$, are the most dominant peaks, but the amplification of the response is primarily driven by the frequency at 9.4 Hz, corresponding to $5f_{wp}$, where $f_{wp} \approx 1.9$Hz. As the fundamental vertical bending frequency for Bridge 1 is 10.5 Hz, this is in close proximity to $5f_{wp}$ resulting in the increase in the displacement response. In Figure A14c and the FFT shown in Figure A14d for Bridge 2, the second peak is suppressed but amplification occurs at frequencies of $3f_{wp}$, $4f_{wp}$, $5f_{wp}$ and $9f_{wp}$. The most significant amplification occurs at $3f_{wp}$ (5.7 Hz) as this is in close proximity to the bridge vertical frequency of 5.3 Hz. For Bridge 3 (Figure A14e,f) the response shows that amplification is driven by the higher integer multiples of $f_{wp}$. Although the first two peaks corresponding to $1f_{wp}$ and $2f_{wp}$ are the dominant frequencies, there is negligible amplification at these frequencies, as the bridge frequency of 14 Hz is not being excited by any of the lower multiples of $f_{wp}$. Bridge 4, which has a vertical bending frequency of 6.8 Hz, is shown to be excited by $4f_{wp}$. The response for Bridge 5, depicted in Figure A14i,j, shows that the low multiples of $f_{wp}$ have little effect on dynamic amplification and only the frequencies at $7f_{wp}$ and $9f_{wp}$ show amplification. In Figure A14k,l the primary wagon pass frequencies $f_{wp}$ and $2f_{wp}$ are suppressed with amplification being evident at $3f_{wp}$, $4f_{wp}$, $5f_{wp}$ and $9f_{wp}$.

### 4.2. Bridge Dynamic Response at 100 km/h—Train DHP-T5

The results for the bridge responses for train DHP-T5 are shown in Appendix B Figure A15. As this train has the longest wagon length, 16.7 m, the primary wagon pass frequency at this speed and for integer multiples of $j < 6$ does not coincide with any of the bridge resonant frequencies. Only in Bridge 1, at $7f_{wp}$, where this is in close proximity of its vertical bending frequency at 10.5 Hz, there is amplification of the response, which is evidenced in Figure A15a. For all of the bridges, the results show that the primary wagon pass frequency, $f_{wp}$, is the dominant frequency for this train and there is very little difference between the quasi-static and dynamic response.

### 4.3. Bridge Dynamic Response at 100 km/h—Train HF-T7

The results for train HF-T7, shown in Appendix B Figure A16, also show similar trends to train DHP-T5, with the response at higher integer multiples of $f_{wp}$ being suppressed. For this train the primary wagon pass frequency is the most dominant and there is little difference in the response between the quasi-static and the dynamic analysis. The displacement response does show the amplification for Bridges 1, 2, 3 and 4 (Figure A16a–c,i), but these are driven by the higher integer multiples of $f_{wp}$. Both trains DHP-T5 and HF-T7 display similar results as their wagon lengths, 16.7 m and 15.7 m, respectively, are considerably longer than the other wagons (see Table 3).

### 4.4. Bridge Dynamic Response at 100 km/h—Train HF-T8

The heavy freight train HF-T8 has the shortest wagon length ($L_w = 5.5$m) and, therefore, results in a higher wagon pass frequency, $f_{wp} = 3$Hz, when compared to the other trains. Furthermore, as $L_w$ becomes closer to the coupling distance between wagons, this leads to narrower frequency peaks which are closer to the integer multiples of the primary wagon pass frequency, $f_{wp}$, as shown in Appendix B Figure A17. The results show that where the second multiple of the wagon pass frequency, $2f_{wp}$, is in close proximity with

the bridge frequency, this causes the primary wagon pass frequency to be suppressed, as shown in Figure A17d,h,l, for Bridges 2, 4 and 6. For these bridges, the displacement response clearly shows the onset of a resonant response, as shown in Figure A17c,g,k. At the train speed of 100 km/h, the integer multiple $2f_{wp}$ (6.1 Hz) is in close proximity to the vertical bending frequency of vibration for Bridges 2 (5.3 Hz), 4 (6.8 Hz) and 6 (5.5 Hz), as is shown in the FFT response in Figure A17d,h,l). The dynamic amplification for these bridges is primarily driven by the frequencies at the higher integer multiples, $2f_{wp}$ and $3f_{wp}$, as shown in Figure A17d,h,l.

For Bridges 1, 3 and 5 the dominant frequency is the primary wagon pass frequency. However, the displacement response for Bridge 1 (Figure A17a) shows signs of a resonance response caused by $3f_{wp}$ (9.2 Hz), which is in close proximity to the vertical frequency of vibration, 10.5 Hz.

For this train type, integer multiples of $f_{wp}$ and where these concide with the natural frequency of vibration of the bridge, can cause the amplification of the bridge response. However, at the higher frequencies, displacements are lower and therefore these may not cause the same level of fatigue damage for frequencies at $1f_{wp}$ or $2f_{wp}$, where these coincide with the bridge frequency. Therefore, to control or limit dynamic amplification, the following steps could be introduced:

-   Have an optimum blanket train speed limit for all train types for a given bridge;
-   Have individual train speeds for different train types for given bridges;
-   For new bridge designs, ensure that the vertical natural frequency vibration is not a factor of the primary wagon pass frequency for the different train types on the route.

The above may not be practical to implement but where there is need to extend fatigue life, or if levels of noise need to be controlled, then these represent the different options that can be adopted.

### 4.5. Train Dominant Frequencies at 100 km/h

Table 4 shows the dominant wagon pass frequencies and the integer multiples of $f_{wp}$ affecting dynamic amplification for each train type. The general pattern which emerges is that trains with shorter wagon lengths (S-T1 and HF-T8) have a much greater impact on the bridge response. For train S-T1, the results summarised in Table 4a show that bridge frequencies in the range of 10–14 Hz have two dominant frequencies, one at the primary wagon pass frequency, $f_{wp}$, and the other at $2f_{wp}$. However, these are shown not to affect dynamic amplification, as this is shown to be influenced by only one or two of the higher integer multiples of $f_{wp}$. For those bridges with frequencies falling between 5 and 7 Hz, only a single dominant frequency exists, but the response amplification is affected by a number of higher integer multiples of $f_{wp}$. Trains DHP-T5 and HF-T7 have the longest wagon lengths. For these trains, the primary wagon pass frequency, $f_{wp}$, is the dominant frequency, but only one of the higher integer multiples is shown to affect dynamic amplification (Table 4b,c). Train HF-T8, which has the shortest wagon length, is shown to have more dominant frequencies as well as more frequencies at higher integer multiples of $f_{wp}$, resulting in the highest effect on dynamic amplification compared to the other trains. Therefore, this train or any trains with short wagon lengths, are more likely to adversely affect fatigue life, as it is more likely that train speeds will coincide with a multiple of the primary wagon pass frequency, $f_{wp}$, leading to higher dynamic amplification. Furthermore, as the results shown in Figure A17k,l, and summarised in Table 4d indicate, they are also more likely to induce higher levels of noise. As shorter wagons are generally not used for passenger trains, then any noise issues would only be of concern to adjacent buildings.

### 4.6. Dynamic Amplification and Critical Speeds

Table 5 presents a summary of the calculated wagon pass frequencies and dynamic amplification at each of the dominant frequencies identified in the FFT plots (Figures A14–A17 in Appendix B) within the range of $j$ = 1–5. The calculated wagon pass frequencies show good agreement with those obtained numerically using Equation (9).

For the studied speed of 100 km/h, the dynamic amplification at the primary wagon pass frequency shows only a modest increase of 2–4% for trains S-T1, DHP-T5 and HF-T7 when compared with the quasi-static analysis for Bridges 1, 3 and 5. For Bridges 2, 4 and 6, the results from the quasi-static analysis are shown to be more conservative. At $j = 2$ the heavy freight train (HF-T8) shows a 3- to 5-fold increase in the dynamic amplification for Bridges 2, 4 and 5. This increase is driven by a resonant condition as the wagon pass frequency, $2f_{wp} = 6.3$ Hz, is now in close proximity to the vertical bending frequency of vibration for these bridges (2, 4 and 5) which are equal to 5.3 Hz, 6.8 Hz and 5.5 Hz, respectively.

**Table 4.** Dominant frequencies at train speeds of 100 km/h.

| **(a) Train S-T1** | | | | |
|---|---|---|---|---|
| **Bridge No.** | **Vertical Bending Frequency** | **Wagon Pass Frequency, $f_{wp}$** | **Dominant Frequencies** | **Frequencies Affecting DAF** |
| | **[Hz]** | **[Hz]** | $j \times f_{wp}$ | $j \times f_{wp}$ |
| 1 | 10.5 | | j = 1 and 2 | j = 5 |
| 2 | 5.3 | | j = 1 | j = 3, 4, 5 and 9 |
| 3 | 14 | | j = 1 and 2 | j = 8 and 9 |
| 4 | 6.8 | 1.92 | j=2 | j = 4, 5 and 9 |
| 5 | 12.1 | | j = 1 and 2 | j = 7 and 8 |
| 6 | 5.5 | | j = 2 | j = 3, 4, 5 and 9 |
| **(b) Train DHP-T5** | | | | |
| **Bridge No.** | **Vertical Bending Frequency** | **Wagon Pass Frequency, $f_{wp}$** | **Dominant Frequencies** | **Frequencies Affecting DAF** |
| | **[Hz]** | **[Hz]** | $j \times f_{wp}$ | $j \times f_{wp}$ |
| 1 | 10.5 | | j = 1 | j = 5 |
| 2 | 5.3 | | j = 1 | - |
| 3 | 14 | | j = 1 | - |
| 4 | 6.8 | 1.41 | j = 1 | - |
| 5 | 12.1 | | j = 1 | j = 10 |
| 6 | 5.5 | | j = 1 | j = 4 |
| **(c) Train HF-T7** | | | | |
| **Bridge No.** | **Vertical Bending Frequency** | **Wagon Pass Frequency, $f_{wp}$** | **Dominant Frequencies** | **Frequencies Affecting DAF** |
| | **[Hz]** | **[Hz]** | $j \times f_{wp}$ | $j \times f_{wp}$ |
| 1 | 10.5 | | j = 1 | j = 7 |
| 2 | 5.3 | | j = 1 | j = 4 |
| 3 | 14 | | j = 1 | j = 10 |
| 4 | 6.8 | 1.52 | j = 1 | - |
| 5 | 12.1 | | j = 1 | j = 9 |
| 6 | 5.5 | | j = 1 | - |
| **(d) Train HF-T8** | | | | |
| **Bridge No.** | **Vertical Bending Frequency** | **Wagon Pass Frequency, $f_{wp}$** | **Dominant Frequencies** | **Frequencies Affecting DAF** |
| | **[Hz]** | **[Hz]** | $j \times f_{wp}$ | $j \times f_{wp}$ |
| 1 | 10.5 | | j = 1, 2 and 3 | j = 3, 4 and 5 |
| 2 | 5.3 | | j = 1, 2 and 3 | j = 2, 3 and 5 |
| 3 | 14 | | j = 1 | j = 4 |
| 4 | 6.8 | 3.15 | j = 2 and 3 | j = 2 and 3 |
| 5 | 12.1 | | j = 1 | j = 4 |
| 6 | 5.5 | | j = 2 | j = 2 and 3 |

The critical speeds of the trains at which the wagon pass frequencies and their integer multiples coincide with the bridge resonant frequencies are given in Table 6. The highlighted speeds are those which excite the bridge resonance frequency causing a spike in the DAF. The effect of this is shown in the Campbell diagrams for Bridges 1–6, shown in Appendix B Figures A18–A23. A Campbell diagram is a straightforward way to represent the effect of resonance on dynamic amplification. Its use in this case enables the visualisation of the train speed at which resonance occurs and its effect in terms of magnitude,

which in this case is based on the DAF calculated using Equation (8). It also helps to identify which of the integer multiples are contributing to dynamic amplification and the train speeds at which these are likely to occur. By showing the resonance frequency of the bridge on the same plots, represented by the horizontal red lines, the train speed at which each integer multiple of the wagon pass frequency (represented by the diagonal lines calculated using Equation (9)) crosses the bridge frequency line indicates when a resonant response is likely to occur. To demonstrate this approach, each of the BS-5400 trains were traversed in 1 km/h increments up to a speed of 300 km/h over each of the analysed bridges. The latter speed is well above the operational speed but the purpose here is to show the application of the Campbell diagram.

**Table 5.** Wagon passing frequencies and dynamic amplification—100 km/h.

| Bridge No. Span and Frequency | BS5400 Train | Wagon Pass Frequency $f_{wp}$—[Hz] | | Dynamic Amplification Factor (DAF) at Dominant Train Load Frequencies | | | | |
|---|---|---|---|---|---|---|---|---|
| | | Calculated | EBB Model FFT | j = 1 | j = 2 | j = 3 | j = 4 | j = 5 |
| 1 $L = 8.84$ m, $f_n = 10.5$ Hz | S-T1 | 1.92 | 1.93 | 1.03 | 0.87 | - | - | 2.79 |
| | DHP-T5 | 1.41 | 1.35 | 1.04 | 0.98 | - | - | - |
| | HF-T7 | 1.52 | 1.45 | 1.03 | 0.96 | - | - | - |
| | HF-T8 | 3.15 | 3.12 | 0.95 | 0.61 | 2.22 | 4.40 | 2.99 |
| 2 $L = 18.1$ m, $f_n = 5.3$ Hz | S-T1 | 1.92 | 1.86 | 0.87 | 0.13 | 8.50 | 3.08 | 2.64 |
| | DHP-T5 | 1.41 | 1.41 | 0.98 | 0.79 | - | - | - |
| | HF-T7 | 1.52 | 1.52 | 0.95 | 0.68 | 1.52 | 6.02 | - |
| | HF-T8 | 3.15 | 3.12 | 0.59 | 5.26 | 2.67 | - | 2.27 |
| 3 $L = 9.3$ m, $f_n = 14$ Hz | S-T1 | 1.92 | 1.93 | 1.02 | 0.90 | - | - | - |
| | DHP-T5 | 1.41 | 1.35 | 1.02 | 0.98 | - | - | - |
| | HF-T7 | 1.52 | 1.45 | 1.02 | 0.97 | - | - | - |
| | HF-T8 | 3.15 | 3.12 | 0.96 | 0.87 | 0.32 | - | 5.94 |
| 4 $L = 21.33$ m, $f_n = 6.8$ Hz | S-T1 | 1.92 | 1.84 | 0.77 | 0.59 | - | 6.31 | 3.20 |
| | DHP-T5 | 1.41 | 1.39 | 0.96 | 0.91 | - | - | - |
| | HF-T7 | 1.52 | 1.50 | 0.94 | 0.88 | - | - | - |
| | HF-T8 | 3.15 | 3.07 | 0.86 | 3.51 | 3.37 | | 2.38 |
| 5 $L = 8.1$ m, $f_n = 12.1$ Hz | S-T1 | 1.92 | 1.94 | 1.03 | 0.92 | - | - | - |
| | DHP-T5 | 1.41 | 1.36 | 1.04 | 1.00 | 0.89 | - | - |
| | HF-T7 | 1.52 | 1.46 | 1.04 | 0.99 | 0.81 | - | - |
| | HF-T8 | 3.15 | 3.14 | 0.97 | 0.82 | 0.45 | 7.77 | - |
| 6 $L = 21.26$ m, $f_n = 5.5$ Hz | S-T1 | 1.92 | 1.84 | 0.75 | 0.16 | 14.58 | 3.28 | 2.66 |
| | DHP-T5 | 1.41 | 1.39 | 0.95 | 0.76 | 0.20 | 17.17 | - |
| | HF-T7 | 1.52 | 1.50 | 0.93 | 0.68 | - | - | - |
| | HF-T8 | 3.15 | 3.07 | 0.75 | 5.87 | 2.67 | - | 2.89 |

The train critical speeds, given in Table 6, have been calculated using Equation (10) for each train type. Although the primary critical speed, at *j* = 1, is well above the operating train speeds, for values of *j* > 2, lower operating train speeds which fall within the operating train speed envelope will cause dynamic amplification at these integer multiples as shown in this work.

In Figures A18–A23, the Campbell diagrams for Bridges 1–6 are shown with the wagon pass frequencies curves for *j* = 1–10, for each of the BS-5400 medium trains. In each of the response plots, the DAF, in general, is shown to increase with train speed as would be expected, showing spikes around those speeds where the wagon pass frequencies, and in some cases, where the integer multiples of $f_{wp}$, coincide with the bridge's natural frequency.

In Figure A18a, for Bridge 1 with train S-T1, the results show a significant spike at a speed of approximately 280 km/h. This speed excites the bridge resonant frequency at 5.3 Hz as the primary wagon pass frequency, at *j* = 1, coincides with this frequency. The response also shows that at certain speed ranges, the DAF estimated by the railway assessment code is conservative (Figure A18c,d for trains HF-T7 and HF-T8). For train HF-T8, a resonance condition can be seen at 170 km/h at the second integer multiple of $f_{wp}$ (Figure A18d).

In Figure A19a,b, for Bridge 2 with trains S-T1 and DHP-T5, the DAFs generally follow the DAF predicted by the assessment code for speeds up to about 120 km/h. For the heavy freight trains (HF-7 and HF-T8) in Figure A19c,d, respectively, two resonance conditions can be seen. For train HF-T7, spikes in the DAF occur at 120 km/h and 180 km/h, corresponding to wagon pass frequencies at $j = 3$ and $j = 2$, respectively. For train HF-T8, spikes in the DAF occur at 84 km/h and 168 km/h, corresponding to wagon pass frequencies at $j = 2$ and $j = 1$, respectively. For these trains, although the primary wagon pass frequency, at $j = 1$, is outside of the speed range of the trains, its multiples are shown to have detrimental impact on dynamic amplification within the train operating speed range. Operating the trains at these speeds can therefore adversely impact the fatigue damage accumulation rate on the bridge.

**Table 6.** Train critical speeds.

| Bridge No. Span and Frequency | BS5400 Train | Train Critical Speeds, $V_{critical}$—[km/h] | | | | |
|---|---|---|---|---|---|---|
| | | $j = 1$ | $j = 2$ | $j = 3$ | $j = 4$ | $j = 5$ |
| 1 $L = 8.84$ m, $f_n = 10.5$ Hz | S-T1 | 547 | 273 | 182 | 137 | 109 |
| | DHP-T5 | 756 | 378 | 252 | 189 | 151 |
| | HF-T7 | 716 | 358 | 239 | 179 | 143 |
| | HF-T8 | 334 | 167 | 111 | 83 | 67 |
| 2 $L = 18.1$ m, $f_n = 5.3$ Hz | S-T1 | 276 | 138 | 92 | 69 | 55 |
| | DHP-T5 | 382 | 191 | 127 | 95 | 76 |
| | HF-T7 | 361 | 181 | 120 | 90 | 72 |
| | HF-T8 | 168 | 84 | 56 | 42 | 34 |
| 3 $L = 9.3$ m, $f_n = 14$ Hz | S-T1 | 729 | 365 | 243 | 182 | 146 |
| | DHP-T5 | 1008 | 504 | 336 | 252 | 202 |
| | HF-T7 | 955 | 477 | 318 | 239 | 191 |
| | HF-T8 | 445 | 222 | 148 | 111 | 89 |
| 4 $L = 21.33$ m, $f_n = 6.8$ Hz | S-T1 | 354 | 177 | 118 | 89 | 71 |
| | DHP-T5 | 490 | 245 | 163 | 122 | 98 |
| | HF-T7 | 464 | 232 | 155 | 116 | 93 |
| | HF-T8 | 216 | 108 | 72 | 54 | 43 |
| 5 $L = 8.1$ m, $f_n = 12.1$ Hz | S-T1 | 630 | 315 | 210 | 158 | 126 |
| | DHP-T5 | 871 | 436 | 290 | 218 | 174 |
| | HF-T7 | 825 | 413 | 275 | 206 | 165 |
| | HF-T8 | 384 | 192 | 128 | 96 | 77 |
| 6 $L = 21.26$ m, $f_n = 5.5$ Hz | S-T1 | 286 | 143 | 95 | 72 | 57 |
| | DHP-T5 | 396 | 198 | 132 | 99 | 79 |
| | HF-T7 | 375 | 188 | 125 | 94 | 75 |
| | HF-T8 | 175 | 87 | 58 | 44 | 35 |

For Bridge 3, which has a resonant frequency of 14 Hz, the DAFs under dynamic conditions only show a marginal increase than those based on the assessment code (Figure A20). A resonance condition exists for train HF-T8 at 225 km/h (Figure A20d), but this is outside the normal train operating speed range.

Bridge 4 shows a single dominant spike at 180 km/h for train S-T1 (Figure A21a). For all other trains traversed over the bridge, the DAF generally follows the assessment code predictions. The DAF for Bridge 5 (Figure A22) also generally follows the DAF predicted by the assessment code for speeds up to 100 km/h. For speeds between 100 and 200 km/h the assessment code overestimates the DAF for trains S-T1 and DHP-T5 (Figure A22a,b).

In Figure A22, for trains HF-T7 and HF-T8 over Bridge 5, the DAF generally follows that of the assessment code, except for train DHP-T5, which shows a marginal increase from 50 km/h onwards (Figure A22b). The results show a resonance condition at 200 km/h for train HF-T8 which results in a spike in the DAF (Figure A22d). However, this is outside of the operational train speed. The DAF for Bridge 6 is shown in Figure A23. Within the operating speed range of all the trains, the DAF follows those predicted by the assessment code; however, for train S-T1 a resonance condition exists at 145 km/h for $j = 2$ (Figure A23a).

*4.7. Bridge Dynamic Response for Trains with Equal Axle Spacing*

In the calculation of the wagon pass frequencies, the locomotive is shown to have little influence on the results except in the case where the locomotive has the same dimensions as the wagons. The calculations presented thus far were for the BS-5400 trains where the axle spacings and coupling distances are variable for each train type. For a hypothetical case where axle spacing and train coupling distances are equal, as illustrated by Figure 6, Equations (17) and (18) are modified to Equations (19) and (20), respectively.

$$f_{wp} = \frac{5jV}{18L_a} \qquad \begin{array}{l} j = 1,2,3 \dots n \text{ integer multiples} \\ L_a = \text{axle spacing} \end{array} \tag{19}$$

$$V_{critical} = \frac{3.6 f_n}{j} L_a \quad f_n = \text{natural frequency of bridge in Hz} \tag{20}$$

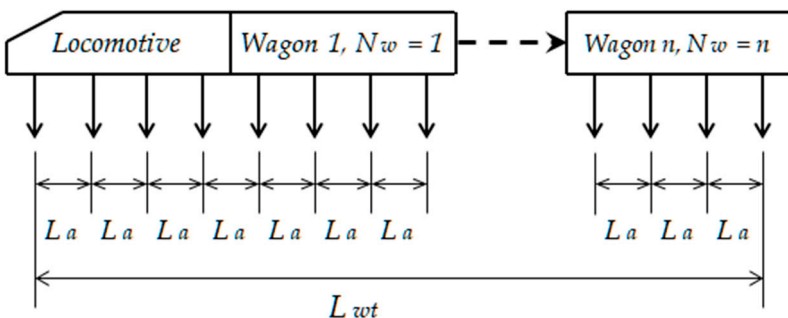

**Figure 6.** Train with equally spaced axles and coupling distance.

To demonstrate this hypothetical train, the axle distance, $L_a$, and the coupling distances, were set to 3.5 m for train HF-T8. The speed of the train (100 km/h) and the axle forces and number of wagons remained unchanged, though the latter is not required in the above equations. The response from these hypothetical trains is shown in Figure A24, displaying a periodic response where the red vertical lines in the FFT plots are the wagon pass frequencies predicted by Equation (19) for *j* = 1 to *j* = 10. The response results show that displacement ranges, shown in Figure A24a,c,e,g,i,k, are significantly reduced compared to the responses for the BS-5400 trains. The frequency peaks are also narrower and more prominent, appearing at the integer multiple of the primary wagon pass frequency. As the primary wagon pass frequency, $f_{wp}$ = 7.9Hz, does not coincide with the natural frequency of vibration of any of the bridges, the peak associated with this frequency is shown to be suppressed in the FFT plots. By banding together the bridge frequencies, those that fall within the 5–7 Hz (Bridges 2, 4 and 6) and those that fall within 10–15 Hz (Bridges 1, 3 and 5), it can be seen that for lower bridge frequencies the primary wagon pass frequency is the most dominant. For the higher bridge frequencies (10–15 Hz), the primary wagon pass frequency becomes suppressed and $2f_{wp}$ becomes the dominant frequency. Although this is a hypothetical train, what the model shows is that having an equally spaced axle and coupling distances helps to reduce the overall displacement ranges that were seen in the responses for the BS-5400 trains. This can have significant benefits when considering fatigue as lower stress ranges generally result in a longer remaining fatigue life.

## 5. Conclusions

This paper has presented the dynamic response analysis of a range of typical short to medium-span plate girder bridges, typically found on the UK railway network, under the actions of the BS-5400 medium traffic mix trains [6]. The purpose of this assessment was to present a simplified mathematical model which can be used to predict bridge dynamic response in the time and frequency domains, as well as identifying how dynamic amplification is affected by wagon pass frequencies at conditions of resonance. The analytical

model to represent the vibration response of the bridge used in this paper is based on the well established Euler–Bernoulli Beam theory for a series of moving loads presented by Fryba [23] and Yang [14]. The model in this study was adapted to allow the assessment of different train configurations in a standardised general form. A closed form solution of the equations was obtained using the inverse Laplace–Carson transform, giving it a distinct advantage in terms of simplicity and efficiency, over other non-closed form solutions using iterative numerical time-step methods.

A general equation for predicting the wagon pass frequency and train critical speeds for a given train configuration was presented and verified using bridge field data and from the frequency response analysis of the model results. The variation in dynamic amplification was compared with that obtained from the UK railway bridge assessment code [14], across a range of train speeds. This was demonstrated using a Campbell diagram which visually showed how dynamic amplification is increased not only at the primary wagon pass frequencies, but also its integer multiples. Although the theory is well established, there is limited work on the actual application of the methodology in assessing the dynamic amplification effects of different train configurations on typical short- to medium-span bridges to date.

The main benefit the model provides is the generalisation of the modelling approach, which makes it easily applicable for the assessment of any bridge and train configuration. As the model is based on the primary bending mode of the bridge, which in most cases is the most damaging for fatigue, the model can accommodate any type of single span simply supported railway bridge where the primary design information, resonant frequency and flexural rigidity are known. Although the model has not considered the train mass effect on the dynamic response of the bridge, which is the subject of a separate investigation by the authors, it provides a comprehensive numerical method by which an initial assessment can be quickly and efficiently made without the need for more complex and time consuming modelling approaches.

The validity of the model was demonstrated by using standard beam theory and FE-based modelling. The modal response and the displacement influence curve were obtained from a 3D FE model of a case study bridge and correlated with the results obtained from the EBB dynamic model. Further validation was provided by comparing independently obtained response data in terms of displacement and accelerations of the case study bridge from actual field measurements and comparing these with the results obtained from the EBB dynamic model. In all cases good correlation was obtained, confirming the predictive capability of the analytical model proposed. The wagon pass frequency calculated using Equation (17) was shown to predict the measured frequencies from the case study bridge successfully. This was subsequently shown to predict the frequency peaks of the integer multiples of the wagon pass frequency from the FFT plot of the dynamic displacement response using the EBB model.

As this work has focused on bridge response at mid-span, using the Campbell diagram the dynamic amplification, which has been based on deflection, reveals the train speeds at which stresses could be amplified. For certain types of trains, it was shown that the amplification is not necessarily dominant at the primary wagon pass frequencies but, instead, at its higher multiples. The DAF suggested by the UK railway bridge assessment code [14] takes into account parameters such as the span of the bridge and the speed of the train. However, it does not take into account any resonance effects or the significance of the wagon passing frequency or its multiples. To show the applicability of the model in assessing dynamic amplification and wagon pass frequencies, the dynamic response of six typical bridges were obtained for a fixed speed of 100 km/h for the BS-5400 medium-mix trains [6]. The following is a summary of the main findings obtained from the analyses carried out:

o    The primary wagon pass frequencies and its integer multiples can cause a significant increase in the dynamic amplification factor when the frequency coincides with the bridge's natural frequency. This condition is more crucial if the primary wagon pass

frequency is a factor of the bridge resonant frequency. These effects are not captured in bridge assessment codes when estimating the DAF.

o   The results show that for longer wagon lengths, the dominant frequency is the primary wagon pass frequency but it is the higher integer multiples which are responsible for dynamic amplification. As the wagon lengths shorten, such as for trains ST-1 and HF-T8, the first, second and third ($j$ = 1, 2, 3) integer multiple of the wagon pass frequency become dominant, but it is the higher integer multiples which are responsible for dynamic amplification. The shorter the wagon, the more higher integer multiples start affecting dynamic amplification.

o   For Bridges 2, 4 and 6, it was found that train HF-T8 had the highest dynamic amplification at the second integer multiple $j$ = 2 of the wagon pass frequency, where this coincided with the bridge natural frequency.

o   Generally, where no resonance conditions prevail, the DAF under dynamic conditions are comparable to those calculated by the design/assessment codes. This was demonstrated using the Campbell diagram for each analysed bridge and train type.

o   The results obtained in this study show that the fatigue damage accumulating on the bridge could potentially be overestimated, or underestimated, if using DAF values based on bridge assessment codes. Where resonance conditions prevail, it is more likely that fatigue damage will be underestimated at train speeds which excite the bridge's resonant frequency.

o   The DAFs obtained from the dynamic analysis for each train type can provide an indication of optimum speed ranges which will minimise the DAF. This can give train operators and bridge asset owners important information which can be used to help prolong bridge fatigue life by specifying operating train speeds for a given train/bridge configuration. Moreover, this work showed that when the train axle spacings and coupling distances become more uniform (equal) the displacement ranges reduce significantly with the frequency content showing distinct narrow peaks. This would also result in reduced stress ranges which is beneficial for fatigue. The design of new trains, where axle and coupling distances are made shorter, can therefore have significant fatigue benefits for bridges.

*Limitiations of the Euler–Bernoulli Beam (EBB) Model and Scope for Further Work*

The model presented was based on the well established Euler–Bernoulli Beam model and provided a closed form solution using the inverse Laplace–Carson transforms. This provides a distinct advantage over existing more complex multi-body system (MBS) and TBI models, in terms of efficiency and simplicity, which are of particular importance for practicing engineers who may not be familiar with the subject matter of the aforementioned models. Most importantly MBS and TBI models require a time-stepping iterative numerical procedure for the system of equations to be solved. The two most common methods in the literature are the Newmark and Runge–Kutta methods. However, these methods can suffer from low convergence rates, especially when dealing with complex systems with high number of degrees of freedom [68]. The methods can also suffer from numerical instability and errors, which are oscillatory in nature, increasing as the time step increases [53].

The proposed model therefore provides a means by which initial assessment of dynamic amplification and the identification of wagon pass frequencies can be made quickly and easily, providing vital information for engineers to make informed decisions on whether further and more detailed investigations are required. In this work the model was extended to include different train configurations in a standardised general form. The BS-5400 standard assessment trains were used for the case studies, but the model verification used real operational train data. This was shown to correlate well with the field data deflection and frequency response.

The proposed model has not included the consideration of the train mass. One of the questions that often arises in moving load problems is whether a moving force or a moving mass model should be used. According to the literature, both the moving load and mass

models often produce similar results, except at resonance conditions, where the moving load model was shown to overestimate the bridge response [48]. The stiffness of the train suspension was only shown to impact the response of the vehicle but had little impact on the response of the bridge. A key indicator whether the mass would impact the response of the bridge is the mass ratio between the train and bridge. For low mass ratios, the bridge response and natural frequency are not significantly impacted, implying that a moving load model would be adequate. For higher mass ratios, it is found that the acceleration at the bridge mid-span can be significantly reduced. This is an area which still needs to be addressed, particularly to understand the impact on fatigue damage, which has not been addressed to the same extent as some of the other areas of study of railway bridge dynamic effects.

The Euler–Bernoulli Beam model, which is the most widely used model, assumes that the cross-sections remain perpendicular to beam axis during deformation. However, for longer slender bridges, or highly flexible structures, the shear deformation may need to be taken into account. In these cases, the Timoshenko beam theory should be adopted. It is possible to extend the model to include a solution using Timoshenko beam theory for bridges spans where this would be more appropriate. However, whether there is any significant benefit in doing this for assessing fatigue damage should be further investigated.

**Author Contributions:** Conceptualisation, A.K.R.; methodology, A.K.R.; software, A.K.R.; validation, A.K.R., and B.I.; formal analysis, A.K.R.; investigation, A.K.R.; resources, A.K.R.; data curation, A.K.R.; writing—original draft preparation, A.K.R.; writing—review and editing, A.K.R., B.I. and D.H.; visualisation, A.K.R., B.I. and D.H.; supervision, A.K.R., B.I. and D.H.; project administration, A.K.R., B.I. and D.H. All authors have read and agreed to the published version of the manuscript.

**Funding:** This research received no external funding.

**Data Availability Statement:** Data are contained within the article.

**Conflicts of Interest:** The authors declare no conflicts of interest.

## Appendix A. EBB Dynamic Model Validation

The validation for the EBB model is carried out using an FE model and in-service response measurements obtained using a 129-year-old U-frame railway bridge, shown in Figure A1a–c, prior to its decommissioning [69]. The bridge is an unballasted short-span, riveted plate girder bridge, which is typically found on the UK railway network. It consists of two main girders and seven transverse cross-girders as shown in Figure A1b. There are two-wheel timbers carrying the tracks which directly sit on the cross-girders. The general side view of the bridge and dimensions are shown in Figure A1c.

*Appendix A.1. Field Measurement Instrumentation*

The field measurements included lateral acceleration measurement on the main girders and vertical displacement measurements on the main and cross-girders and their mid-spans. The acceleration response was measured using three accelerometers and the vertical displacements using an iMetrum Video Gauge. The positions of the accelerometers and camera targets are shown in Figure A2. The measurements of other locations were also recorded; however, only the results from these locations are used in this assessment. The field measurements were obtained under four different but similar passenger trains running over the bridge at low speeds since the bridge was located near a railway station.

The vertical response data was compared with the displacement time history predicted by the EBB model and acceleration data was used to perform an FFT analysis and compare both sets of results in the frequency domain. Although the measured acceleration data is in the lateral direction whilst the EBB model provides acceleration in the vertical direction, the frequency content should be able to identify the wagon pass frequency and this is the focus of the comparisons with the EBB model.

As the EBB dynamic model only predicts the vertical response of the bridge, then the frequency content of the response, through an FFT analysis, can be compared with the FFT of the measured bridge lateral acceleration response. The lateral response of the main girder top flange will be due to the U-frame action of the bridge, which will be a function of the vertical response. Therefore, the frequencies arising from the passing train will be observable in the lateral response FFT, and this can be compared with the FFT of the response obtained from the EBB dynamic model.

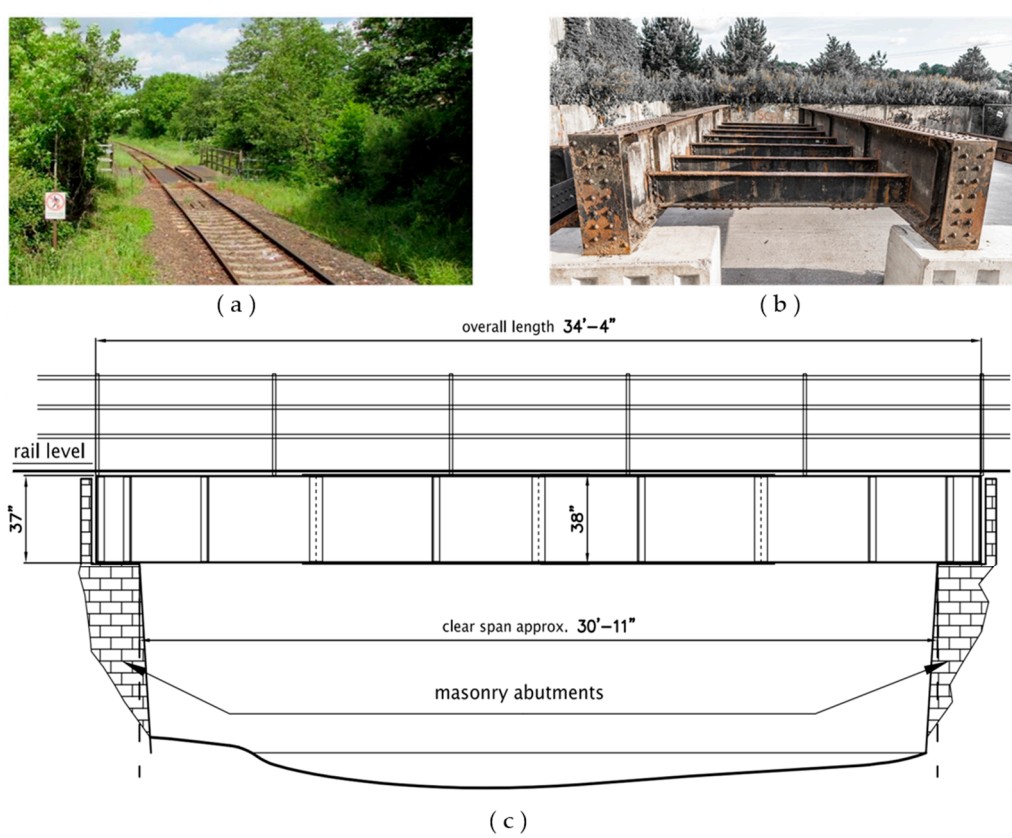

**Figure A1.** Typical U-frame plate girder bridge used for analytical model validation, (**a**) Bridge track view, (**b**) bridge frame main & cross girders, (**c**) bridge side elevation [69].

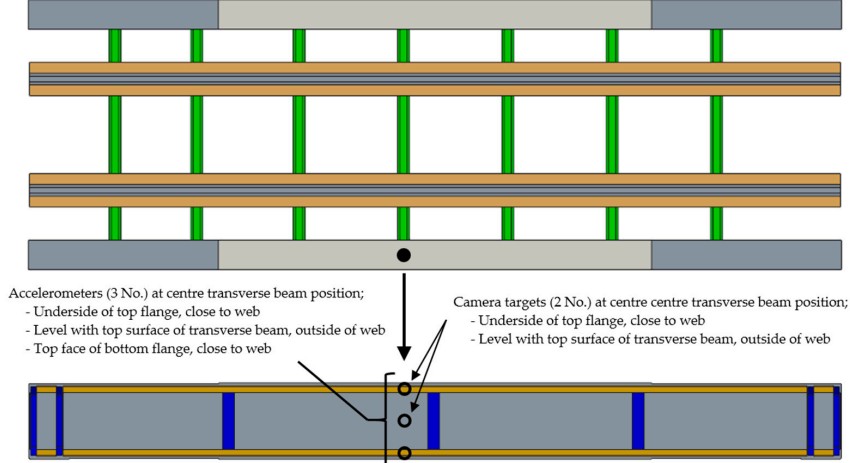

**Figure A2.** Accelerometer and camera target positions on main girder [69].

*Appendix A.2. Finite Element Analysis*

A 3D CAD model of the plate girder bridge was created using Creo Parametric 3.0, with the dimensions shown in Figures A3 and A4. The 3D assembly model of the bridge structure also included the transverse girders, wheel timbers and the rail tracks, as shown in Figure A5a. The model was then imported into the Siemens NX Nastran finite element package to create an FE model of the bridge as shown in Figure A5b. The model was meshed using 20-noded CHEXA solid elements except for the vertical L and T plate stiffeners which were modelled using shell elements. The members were tied to each other via coincident nodes and therefore the local geometry of the connections (i.e., rivets) has not been explicitly modelled. This is deemed acceptable since the global response of the bridge is of interest. The material properties used in the FE model are as summarised in Table A1.

The FE model was analysed for free–free end conditions to simulate a modal hammer impact test performed on the bridge, i.e., hanging the bridge structure at four points using a mobile crane as described in [70]. The frequency obtained by the modal impact test was equal to 51 Hz, corresponding to the first vertical bending mode [70]. The FE analysis for the same free-free condition predicts a frequency of 51.85 Hz, as shown in Figure A6a. This result compares very well with the frequency of 51 Hz and the mode shape predicted by the modal impact test, Figure A6b.

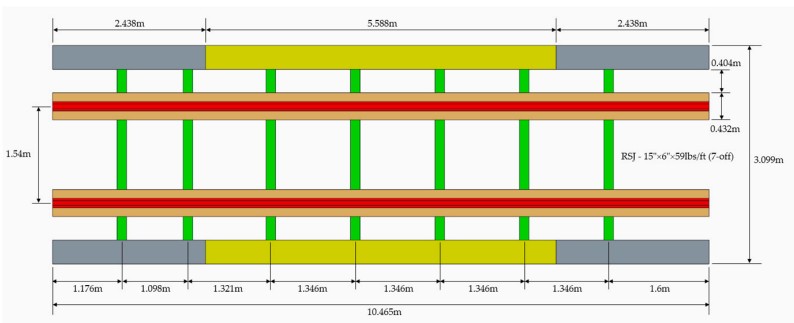

**Figure A3.** Bridge arrangement drawing.

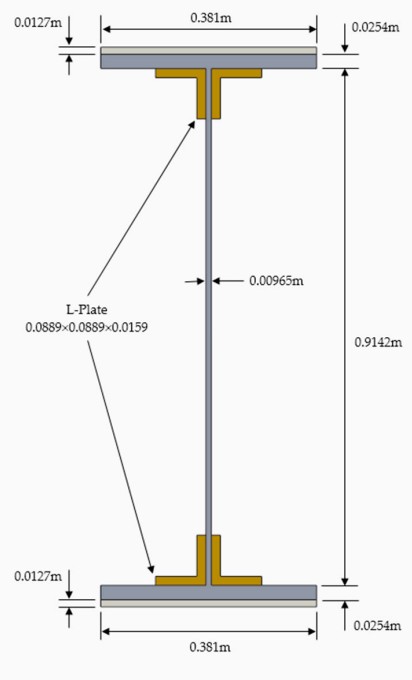

**Figure A4.** Main girder dimensions.

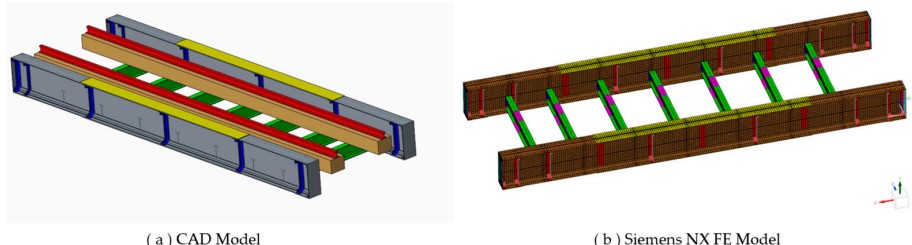

( a ) CAD Model  ( b ) Siemens NX FE Model

**Figure A5.** Three dimensional CAD and FE models of the bridge.

**Table A1.** NX FE model material data.

| Parameter | | |
|---|---|---|
| Bridge main and transverse girders density | 7853 | kg/m$^3$ |
| Rail density | 7800 | kg/m$^3$ |
| Wheel timbers and floors density | 2541 | kg/m$^3$ |
| Young's Modulus, $E_{steel}$ | 190 | GPa |
| Young's Modulus, $E_{wood}$ | 14 | GPa |
| Poisson Ratio, $\nu_{steel}$ | 0.3 | |
| Poisson Ratio, $\nu_{wood}$ | 0.45 | |

Since the Matlab EBB model uses the fundamental vertical bending frequency of the bridge, the purpose of this analysis is to determine this frequency for the whole bridge structure for the simply supported case representing the in-service boundary conditions. By using the material properties summarised in Table A1, modal analysis of the whole bridge structure under simply supported boundary conditions predicted the fundamental vertical bending mode frequency equal to $f_n \approx 20$ Hz (Figure A7).

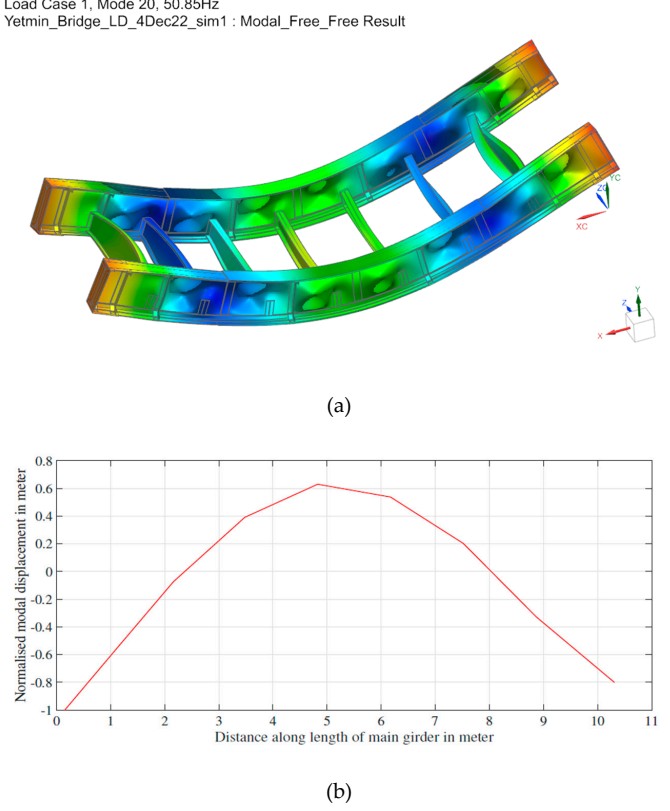

Load Case 1, Mode 20, 50.85Hz
Yetmin_Bridge_LD_4Dec22_sim1 : Modal_Free_Free Result

(a)

(b)

**Figure A6.** Modal analysis correlation with modal impact test, (**a**) Siemens NX free—free modal analyis 51 Hz, (**b**) Modal impact test result—51 Hz (free—free) [70].

Yetmin_Bridge_LD_4Dec22_sim1 : Model_Simp_Supported Result
Load Case 1, Mode 3, 19.60Hz

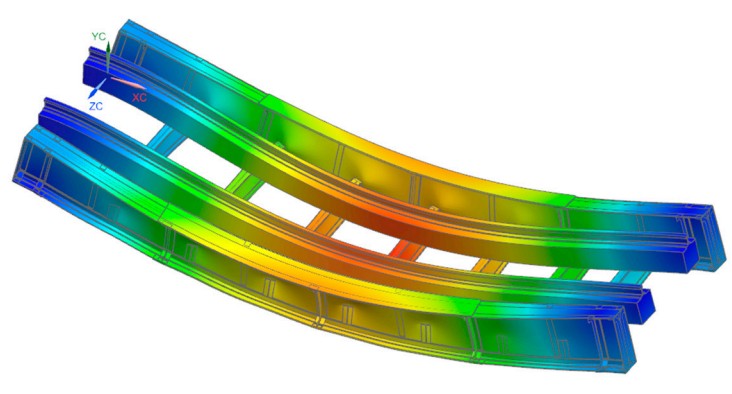

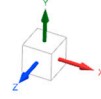

**Figure A7.** Siemens NX FE model modal analysis—simply supported—$f_n \approx 20$ Hz.

*Appendix A.3. Measured Bridge Response Correlation with EBB Dynamic Model*

The bridge data and train types used in the EBB dynamic model are shown in Table A2. The trains used in the EBB dynamic model are shown in Figure A8 and represent train traffic running over the bridge before it was decommissioned. Using the same properties for the bridge given in Table A2 and standard beam theory [23] (Equation (A1)), the EBB dynamic model predicts a frequency of $f_n \approx 20$ Hz for the vertical bending natural frequency of vibration for the simply supported case. This compares very well with the predicted FE analysis frequency of 20 Hz as shown in Figure A7.

**Table A2.** Case study bridge and train data.

| EBB Model Input Data | | |
|---|---|---|
| Span, *L* (between supports) | 9.78 | m |
| Uniformly Distributed Mass *(UDM)*, *μ* | 1748 | kg/m |
| Young's Modulus, *E* | 190 | GPa |
| Second Moment of Area, *I* (at mid-section) | 0.01358 | m$^4$ |
| First Vertical Bending Frequency, $f_n$ | 20 | Hz |
| British Rail Class 158—Diesel Multiple Unit (DMU) | 38 | tons |
| British Rail Class 168—Diesel Multiple Unit (DMU) | 37 | tons |
| British Rail Class 166—Diesel Multiple Unit (DMU) | 37 | tons |

$$f_n = \frac{\pi}{2L^2}\sqrt{\frac{EI}{\mu}} \tag{A1}$$

For the verification of the EBB dynamic model for deflection, the FE model was used to simulate a moving 10 t load traversing the bridge using static analysis to obtain the load–deflection influence curve at mid-span. The influence curve is then compared with the results predicted from the quasi-static and EBB dynamic models resulting in reasonably good correlation, as shown in Figure A9. This shows that the EBB dynamic model is representative of the bridge in terms of stiffness for the first vertical bending mode, which is the primary mode of interest in this study.

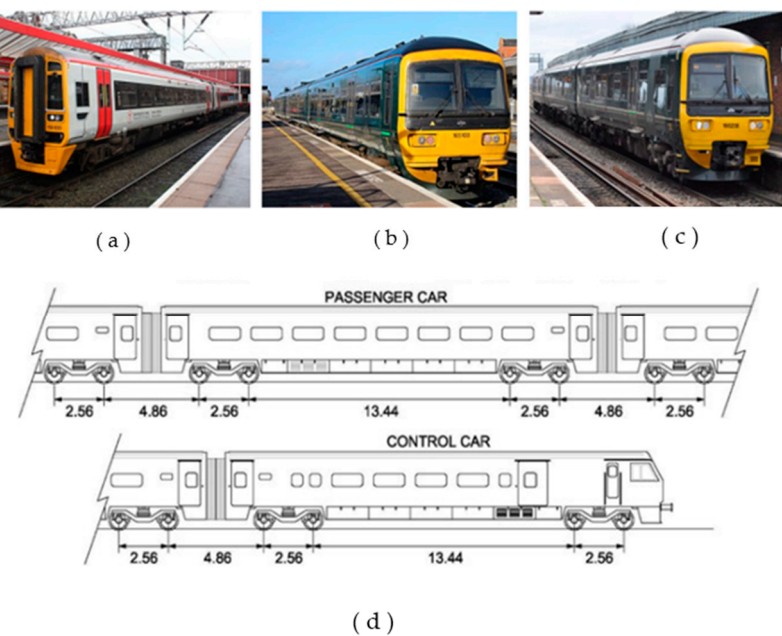

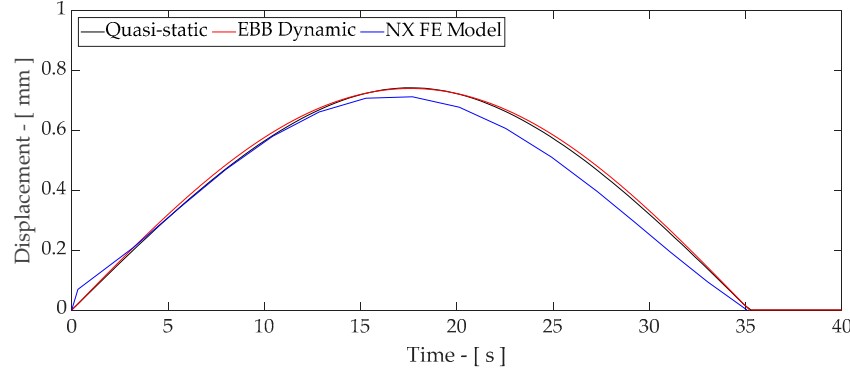

**Figure A8.** British Rail train classes, (**a**) Class 158, (**b**) Class 165, (**c**) Rail Class 166, (**d**) Passenger and contrial car axle spacing [71].

**Figure A9.** Comparison of deflection between EBB model and 3D FE analysis.

As mentioned previously, only the bridge lateral acceleration and vertical responses were measured and these are available at the mid-span section. Therefore, these results can be compared with the EBB dynamic model predictions. The vertical displacement responses can be compared directly. However, the lateral accelerations measured at the underside of the top flange (Figure A2) can be compared with the vertical acceleration response of the bridge for its frequency content. This is due to the U-frame action of the bridge response under the moving load, where the main driving wagon pass frequencies should also be observable in the bridge lateral response measured on the main girder. The comparison can therefore be made by performing FFT of the lateral acceleration response and comparing this with the FFT of the bridge vertical acceleration response. This will enable the wagon pass frequencies to be identified for the given train and compared with the calculated frequencies as predicted by Equation (9).

The EBB dynamic analysis was carried out for the British Rail Class trains shown in Figure A8. The field measured response results are available for four train crossings over the bridge. The train configurations consisted of only one engine car followed by two passenger cars for the first three trains. The fourth train only had one passenger car. The vertical displacement response and the FFT frequency analysis for each train crossing are shown in Figures A10–A18. As it can be seen, the EBB model predicts the frequency as well

as the magnitude of the displacement cycles of the bridge reasonably well, compared to the actual measured response. The FFT frequency analysis shows a single dominant frequency for each train which is the wagon pass frequency. The bridge vertical frequency is not evident in the FFT response as the bridge is not being sufficiently excited by the moving train due to its slow speed. The vertical response of the bridge will become significant if the wagon pass frequency is close to the bridge vertical natural frequency. The wagon pass frequency is a critical property of the response, as this will determine the magnitude of the stress cycles and hence influence the fatigue life of the bridge. The prediction of the wagon pass frequency from the EBB dynamic model FFT response shows very good agreement with the measured frequency, as shown in Table A3, and with the predicted frequency, as calculated by Equation (9).

**Table A3.** Measured frequency response and EBB Matlab model comparison.

| Train | Train Speed [km/h] | Train Configuration | Wagon Pass Frequency, $f_{wp}$—[Hz] | | |
|---|---|---|---|---|---|
| | | | Measured Response FFT | EBB Dynamic Model FFT | Calculated [Equation (9)] |
| 13:48 to Gloucester (Figure A10) | 30 | 1 Locomotive 2 Wagons | 0.388 | 0.369 (−4.9%) | 0.397 (2.3%) |
| 15:37 to Gloucester (Figure A11) | 26.5 | 1 Locomotive 2 Wagons | 0.349 | 0.326 (−6.9%) | 0.351 (0.6%) |
| 13:20 to Weymouth (Figure A12) | 27 | 1 Locomotive 2 Wagons | 0.345 | 0.332 (−3.8%) | 0.357 (3.5%) |
| 16:26 to Weymouth (Figure A13) | 29.5 | 1 Locomotive 1 Wagons | 0.419 | 0.422 (0.7%) | 0.442 (5.5%) |

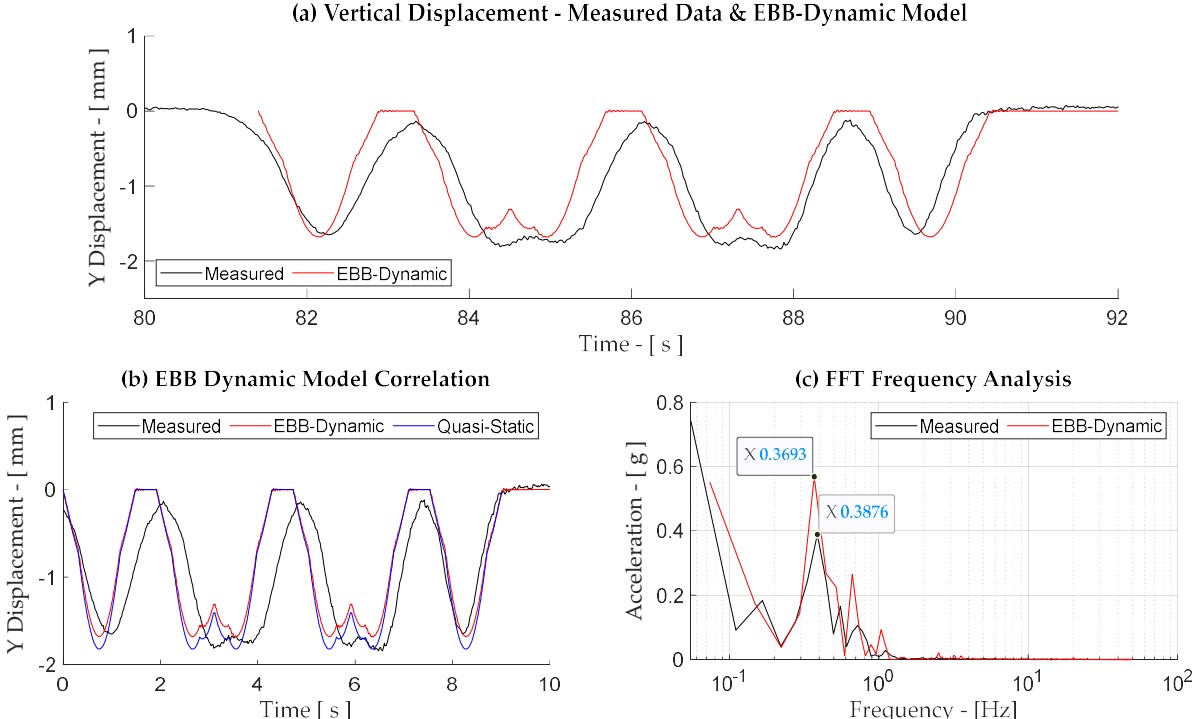

**Figure A10.** Bridge vertical displacement-1348 to Gloucester train.

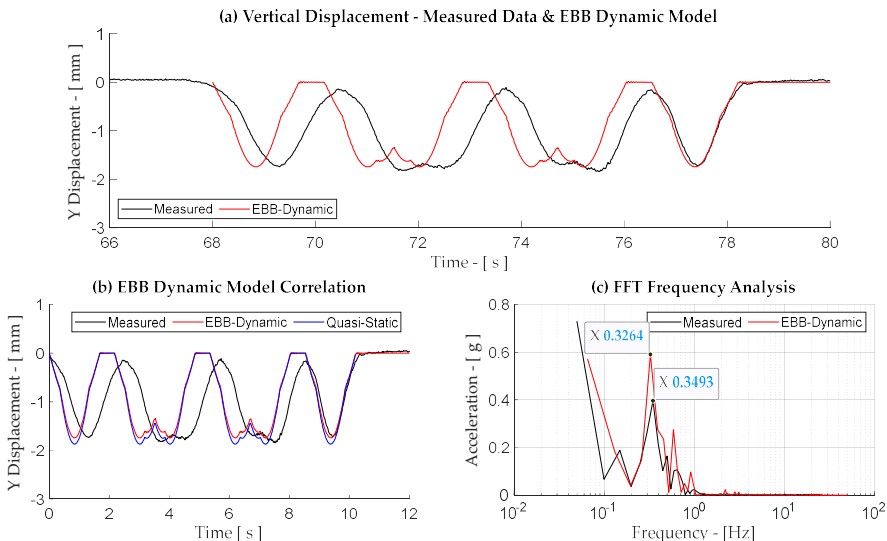

**Figure A11.** Bridge vertical displacement-1537 to Gloucester train.

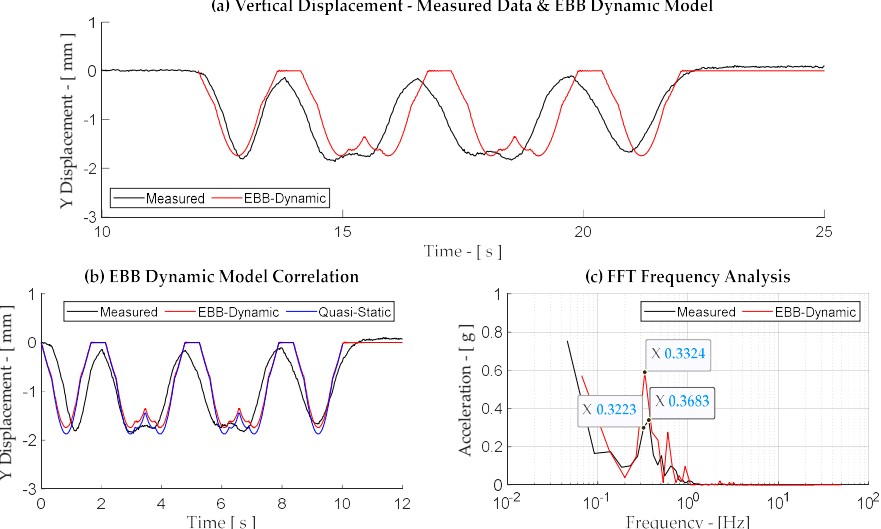

**Figure A12.** Bridge vertical displacement-1320 to Weymouth train.

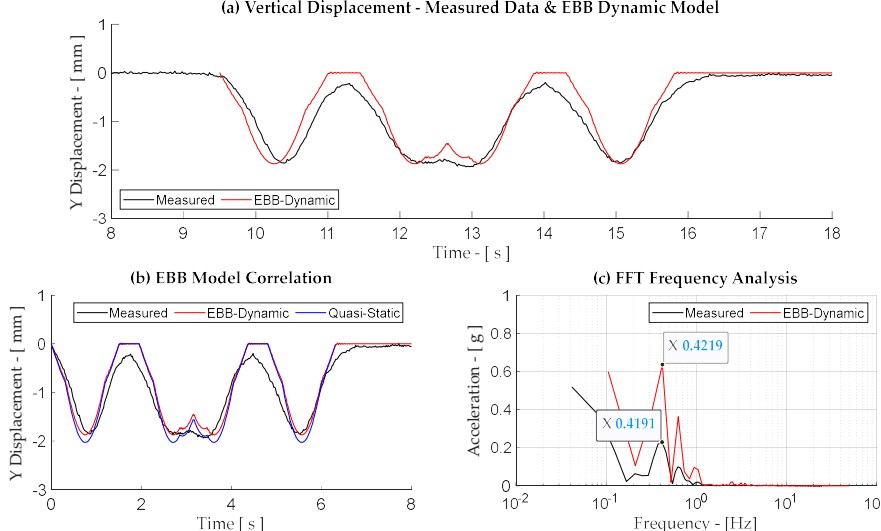

**Figure A13.** Bridge vertical displacement plane 6-1626 to Weymouth train.

## Appendix B. Results for Case Study—Trains S-T1, DHP-T5, HF-T7 and HF-T8

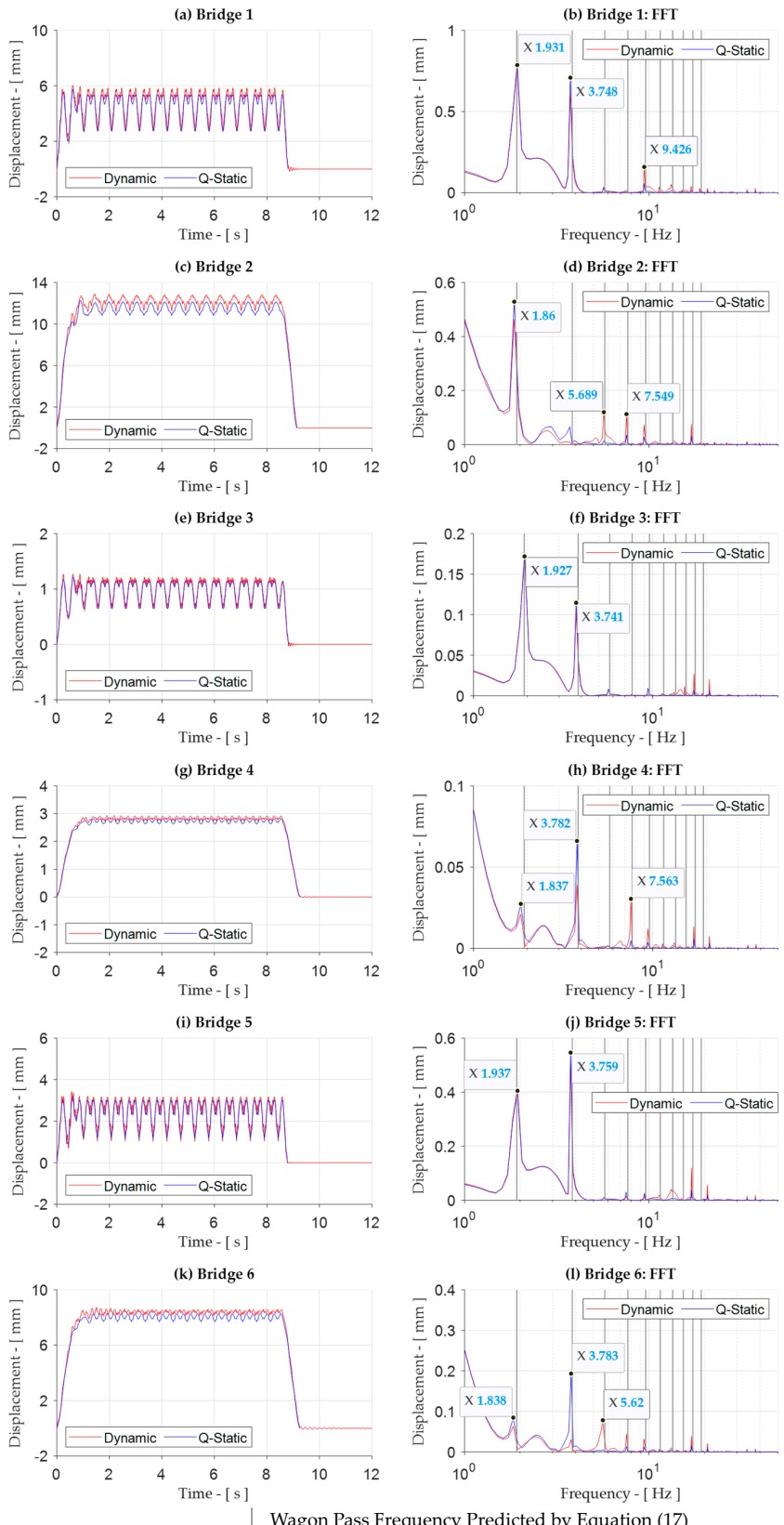

**Figure A14.** Displacement response and FFT for Bridges 1—6: Train S-T1.

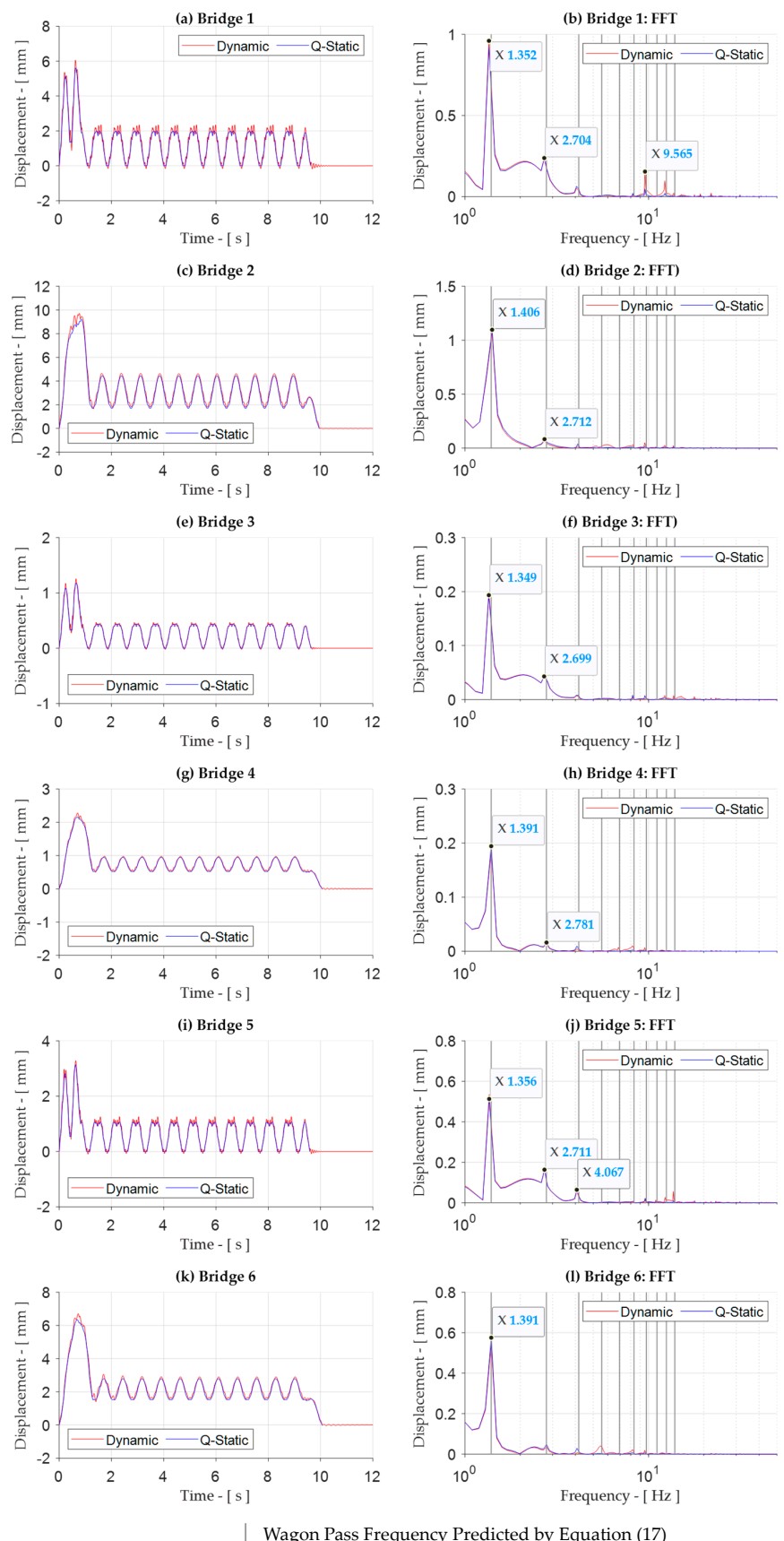

Wagon Pass Frequency Predicted by Equation (17)

**Figure A15.** Displacement response and FFT for Bridges 1—6: Train DHP-T5.

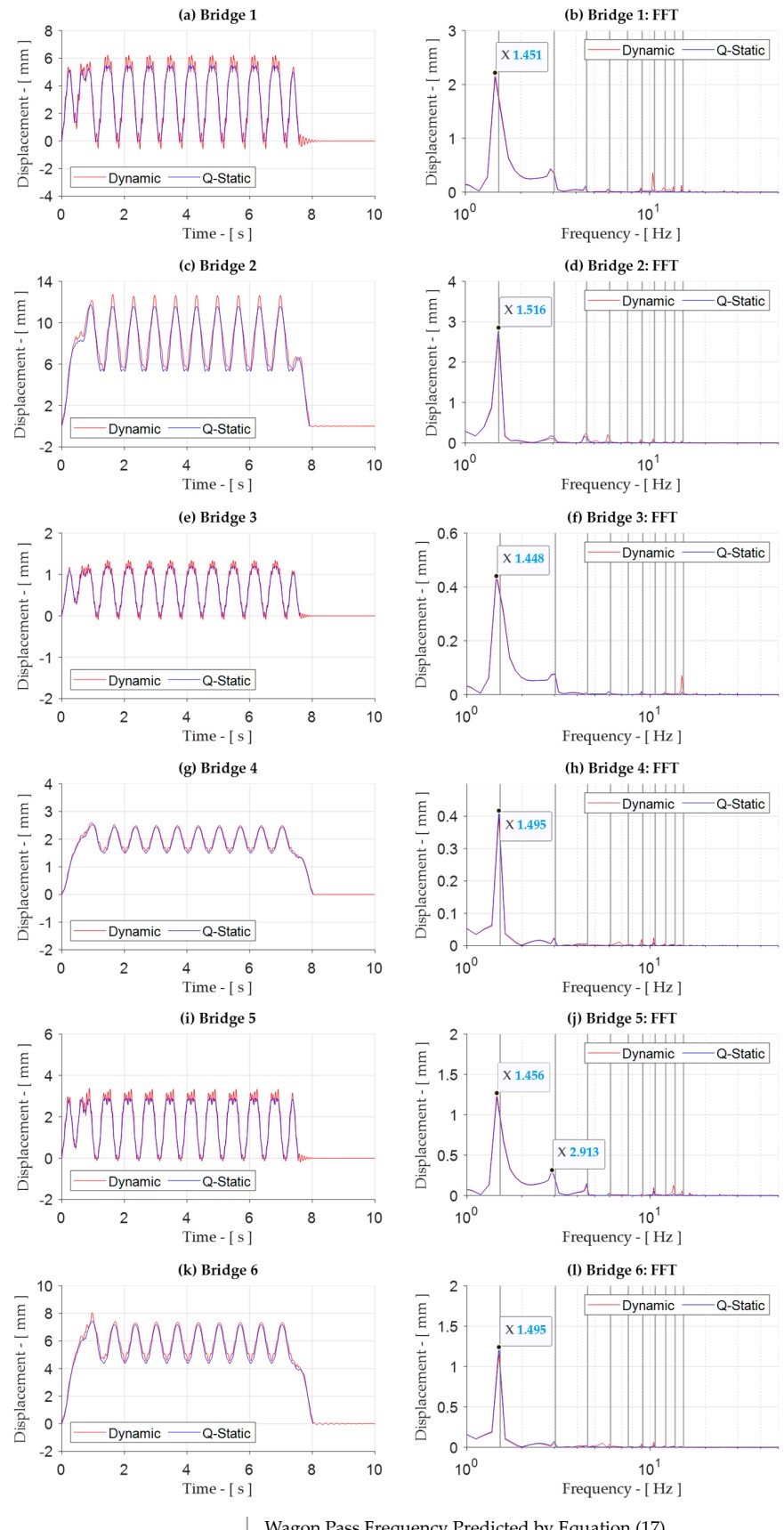

| Wagon Pass Frequency Predicted by Equation (17)

**Figure A16.** Displacement response and FFT for Bridges 1—6: Train HF-T7.

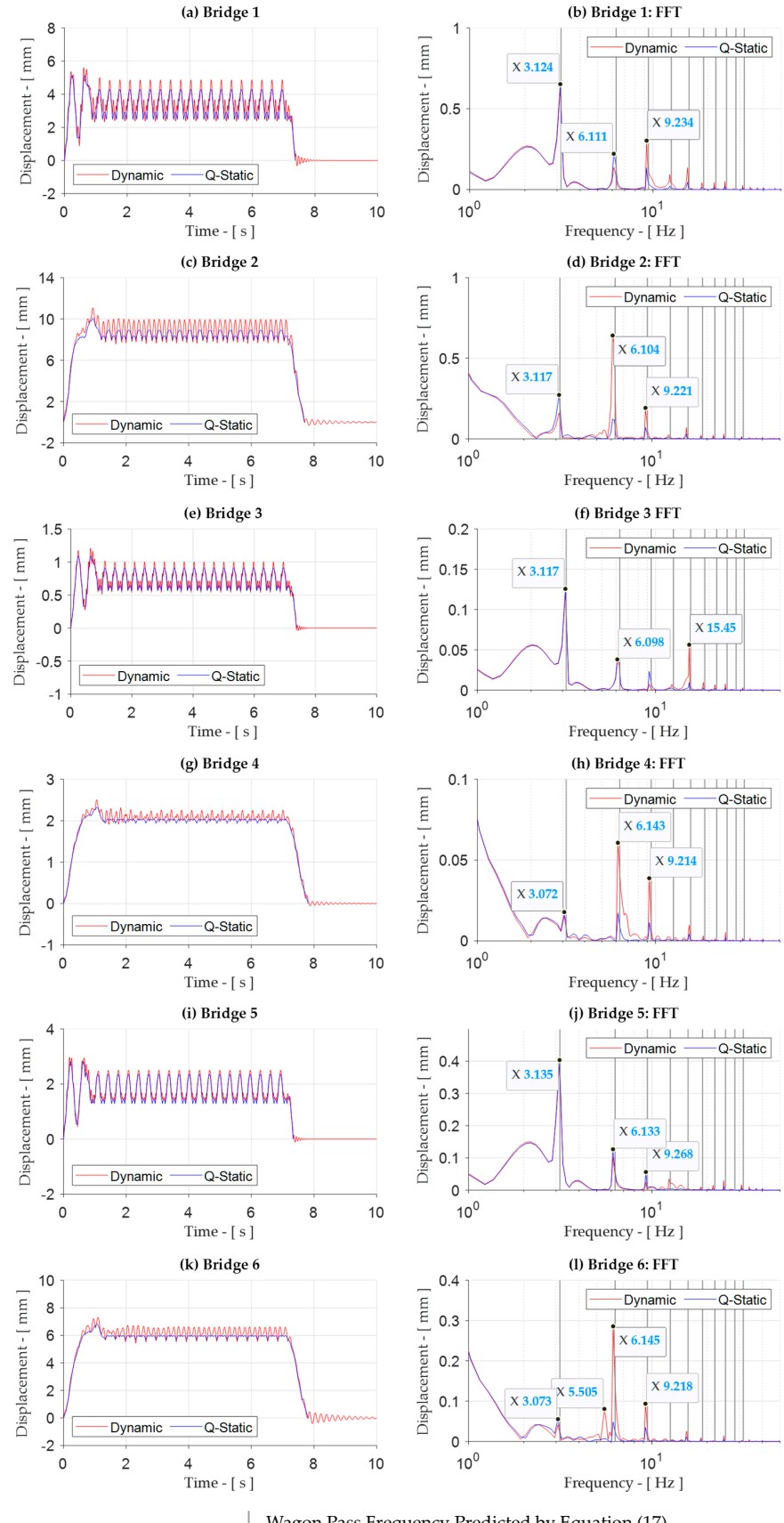

Wagon Pass Frequency Predicted by Equation (17)

**Figure A17.** Displacement response and FFT for Bridges 1—6: Train HF-T8.

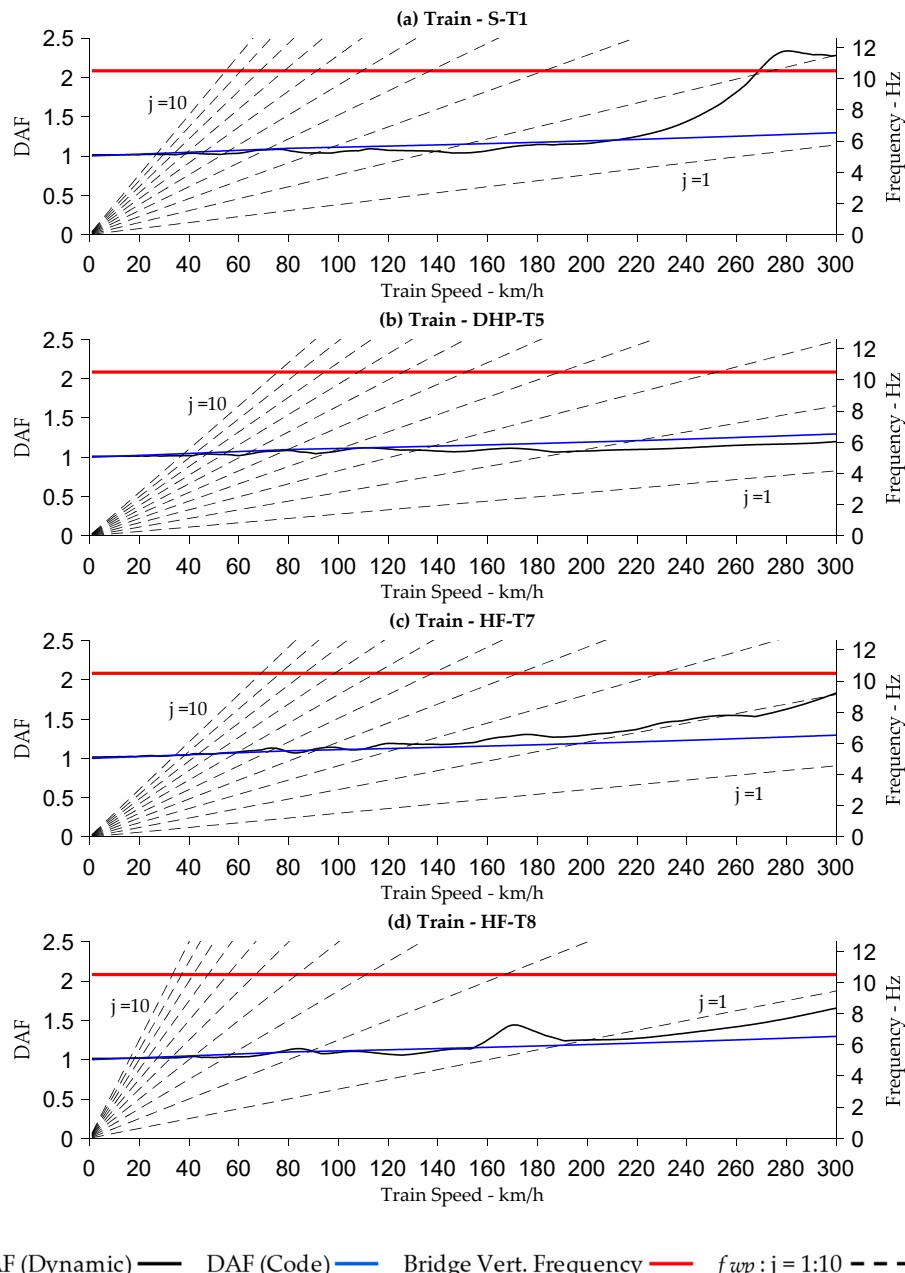

**Figure A18.** Dynamic amplification medium train mix—Bridge 1.

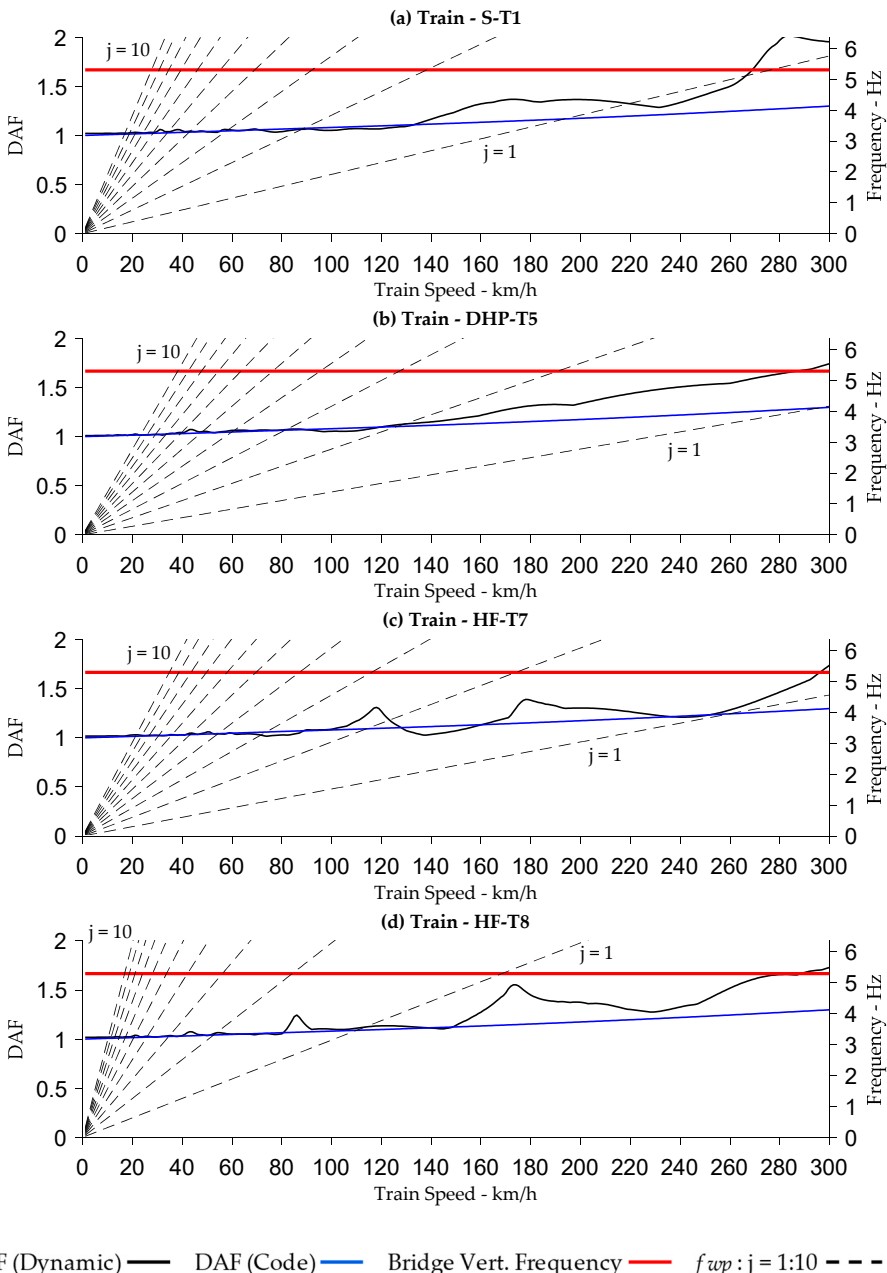

**Figure A19.** Dynamic amplification medium train mix—Bridge 2.

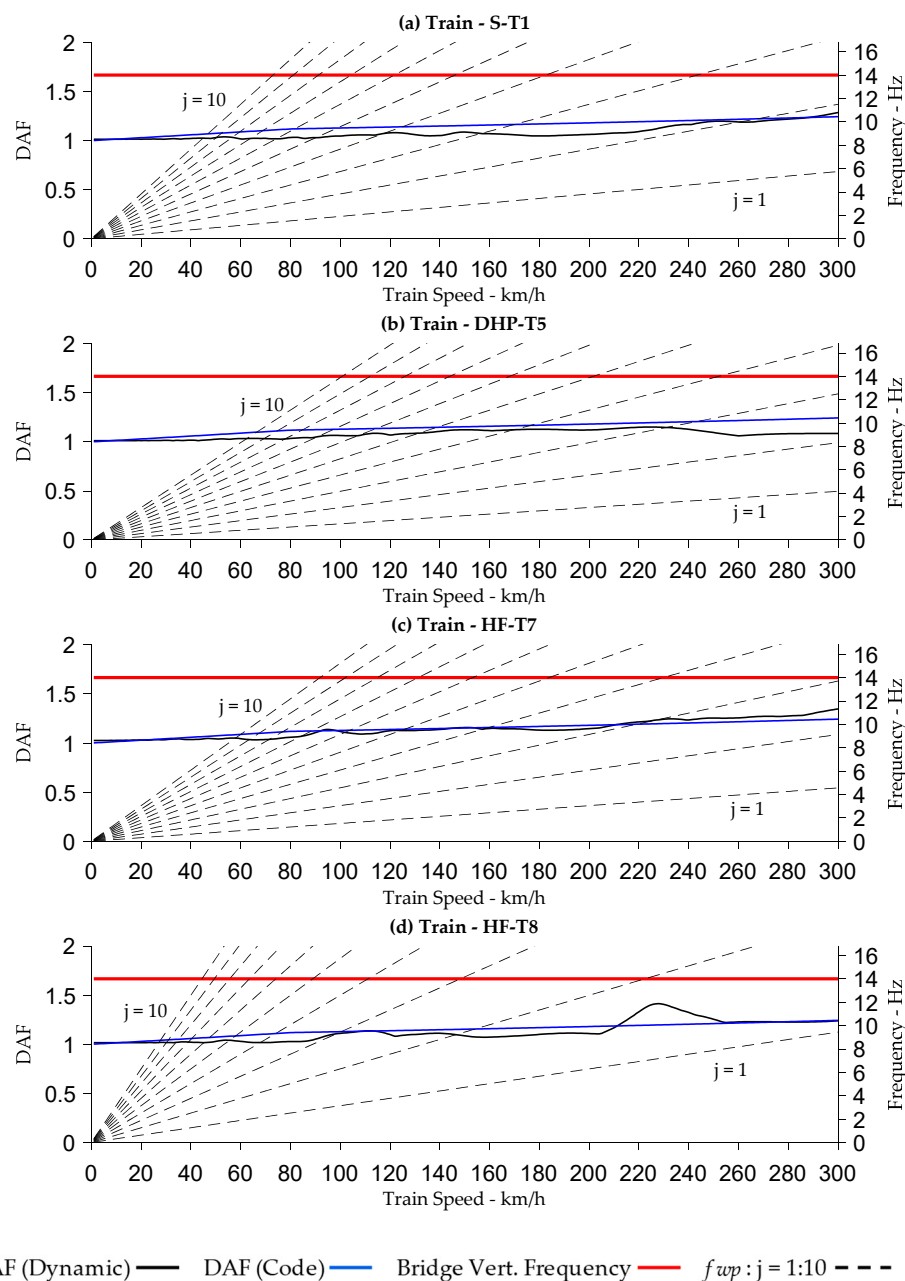

**Figure A20.** Dynamic amplification medium train mix—Bridge 3.

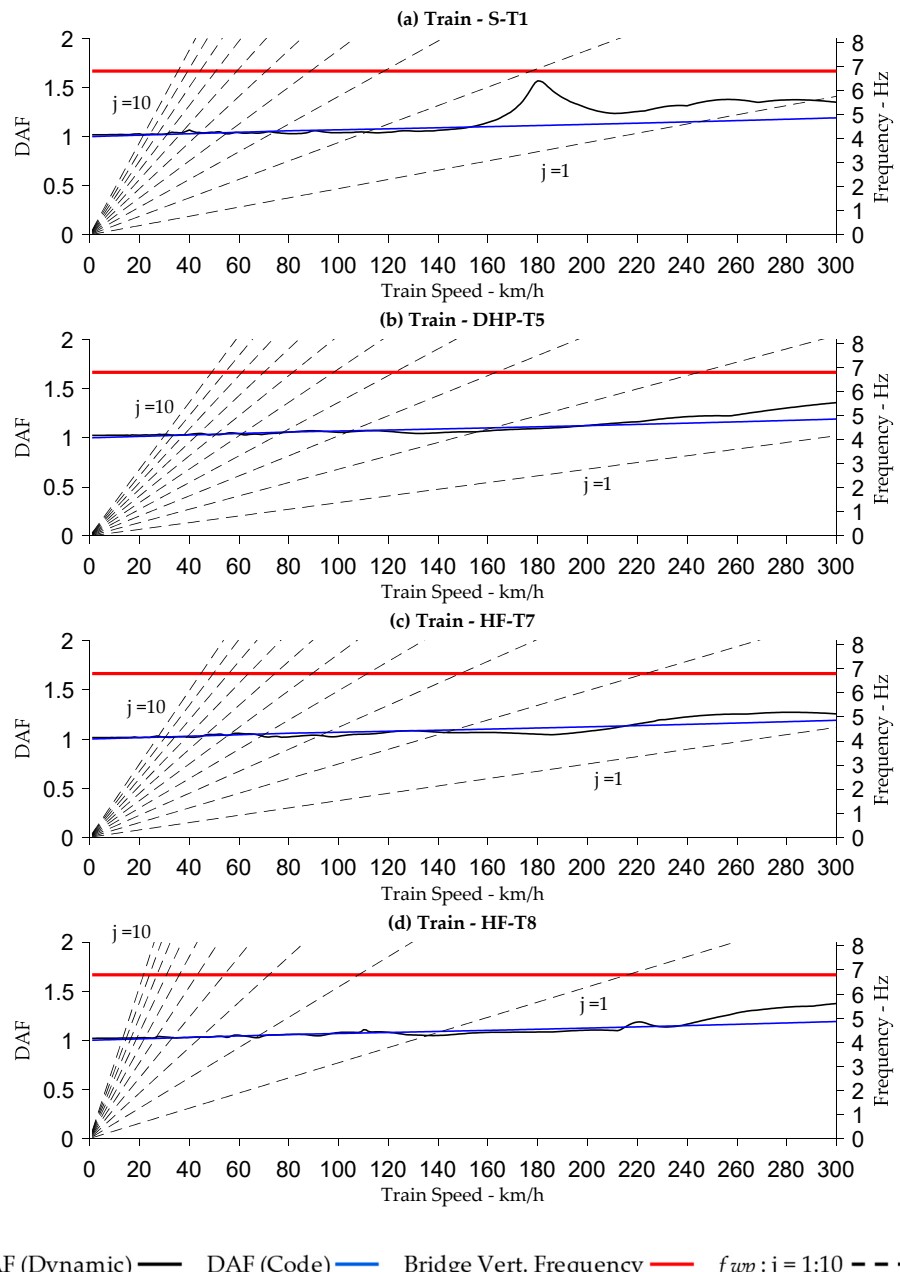

**Figure A21.** Dynamic amplification medium train mix—Bridge 4.

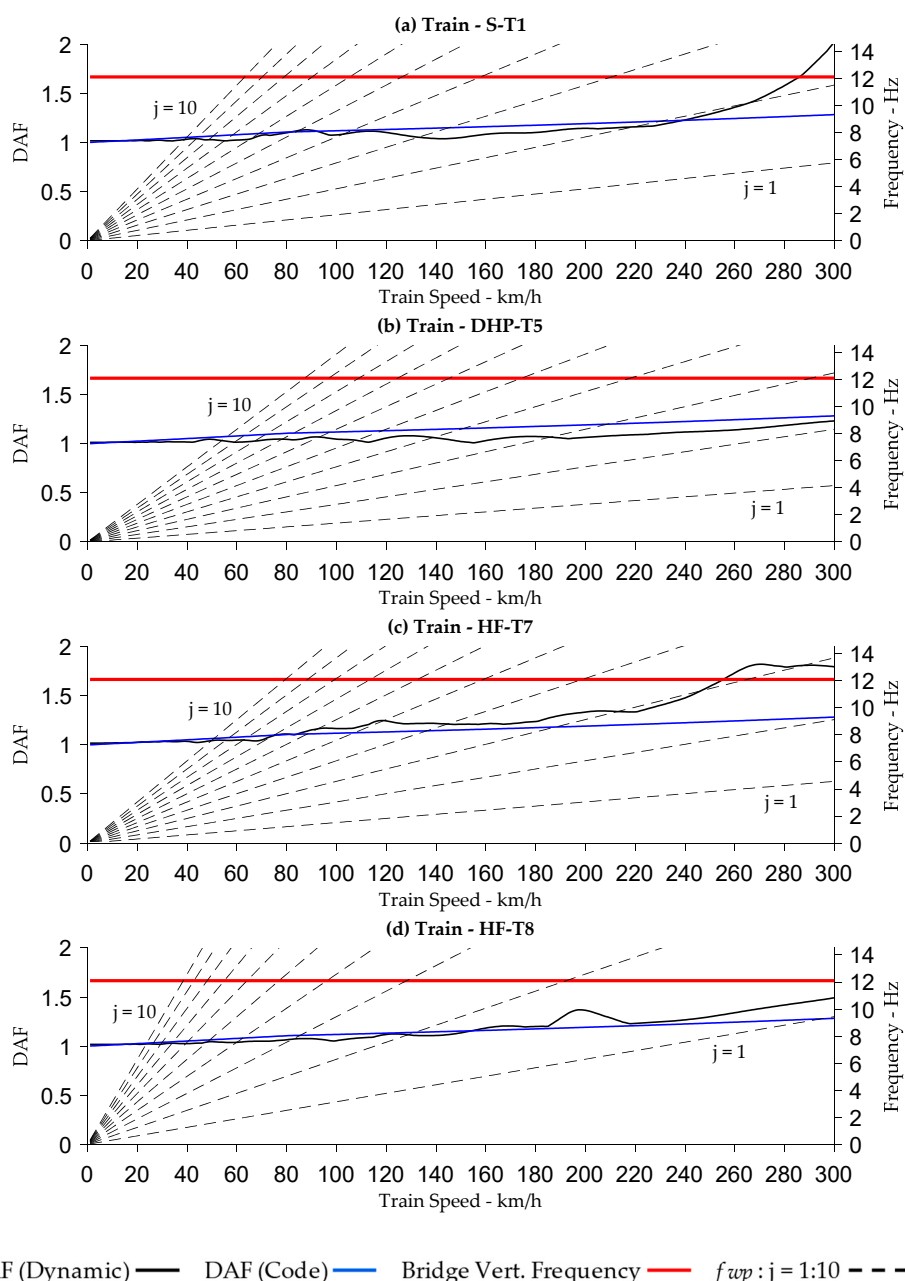

**Figure A22.** Dynamic amplification medium train mix—Bridge 5.

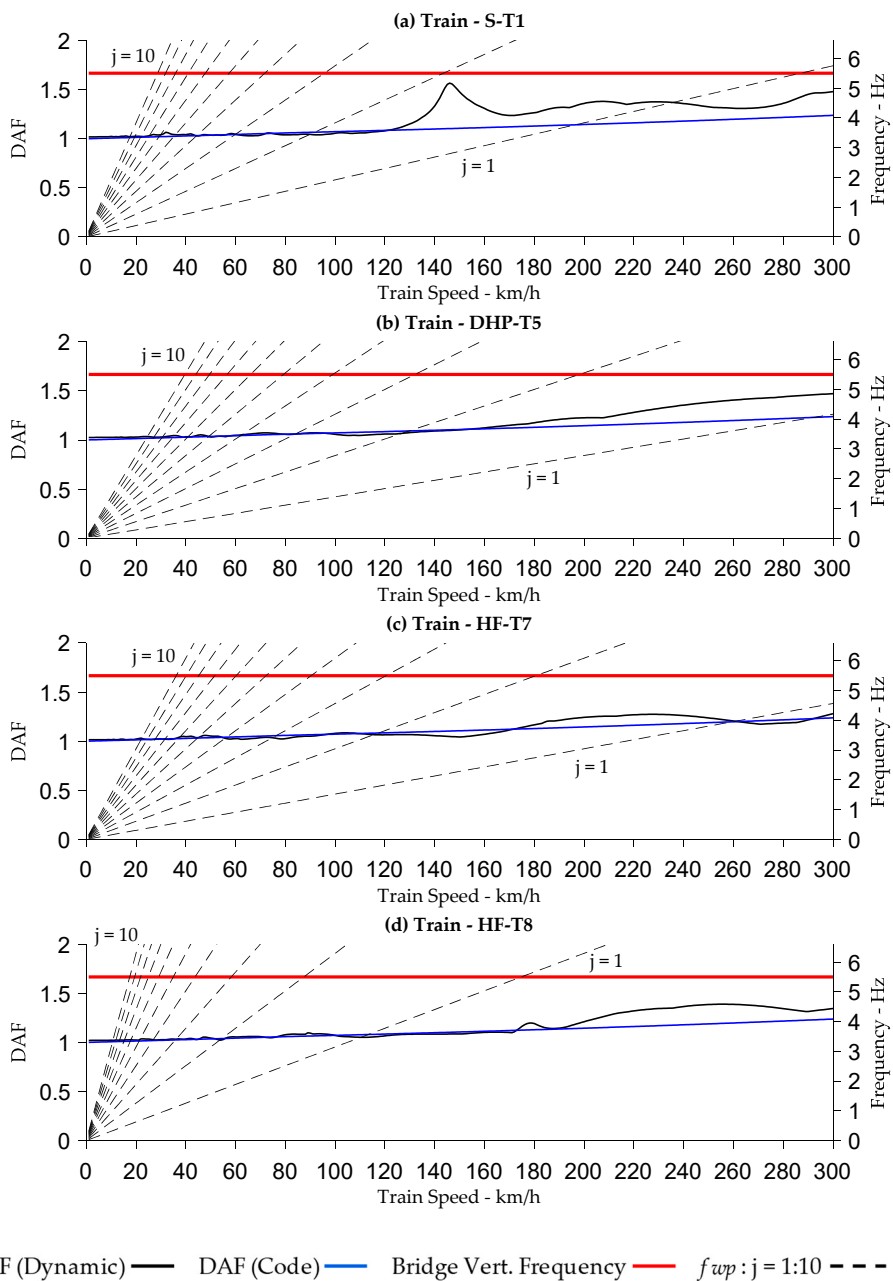

Figure A23. Dynamic amplification medium train mix—Bridge 6.

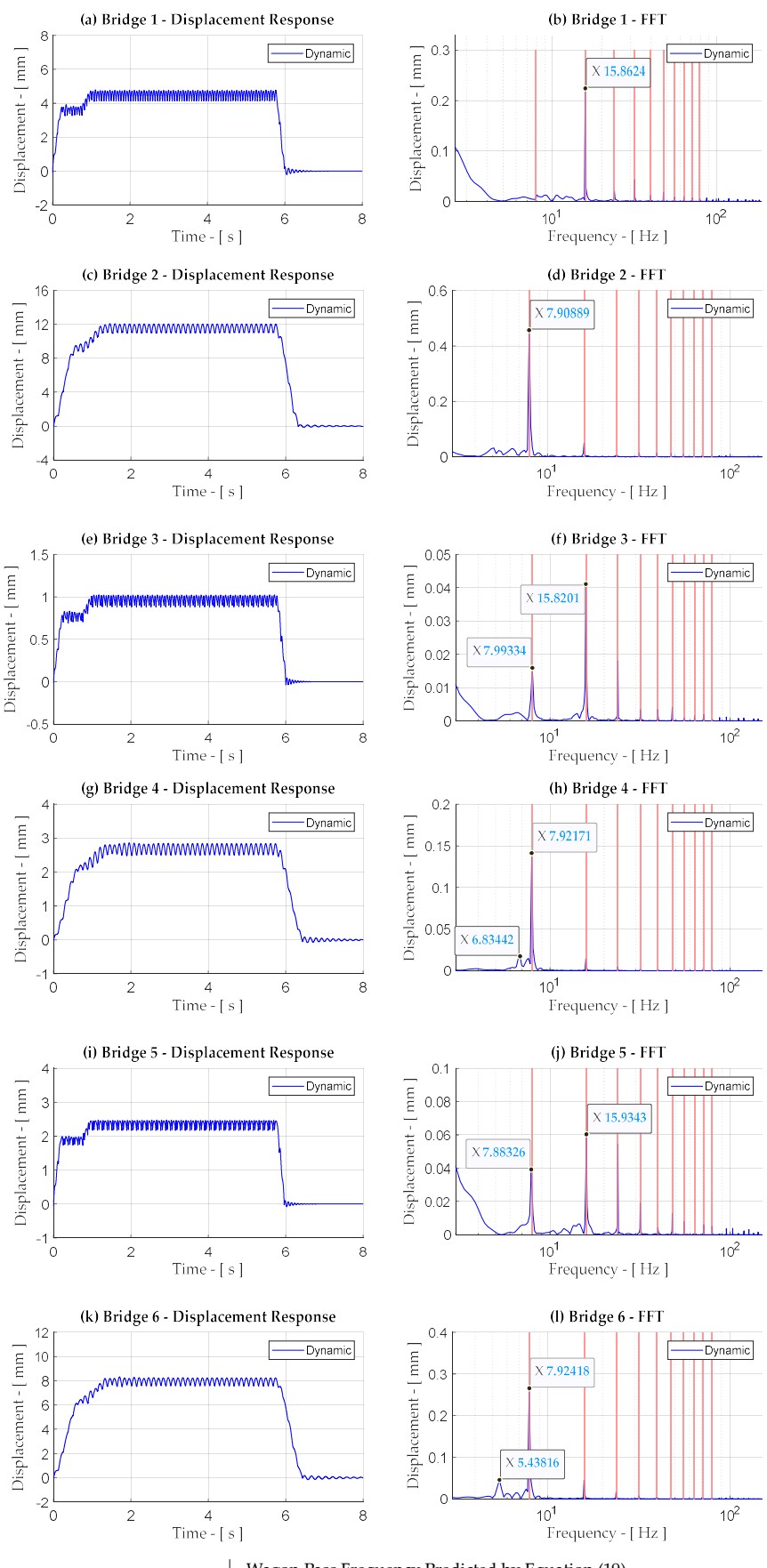

Figure A24. Bridges 16—3.5 m axle and coupling spacing 100 km/h.

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
