# Peer review of "Dynamic Amplification of Railway Bridges under Varying Wagon Pass Frequencies"

_infrastructures, doi:10.3390/infrastructures9030062_

Round 1

Reviewer 1 Report

Comments and Suggestions for Authors

The referred manuscript presents a modified EBB model to predict the dynamic response of girder bridges under the load of passing trains, which has been validated by an FEA model and field-measured data. The good correlation shows that the model could successfully help assess bridge damage. Using the proposed model, the effect of pass frequencies on the resonance of different bridges and trains is discussed. The paper is well-written and structured. The research work is thoroughly discussed. 

However, the manuscript needs to be shorter for publication in a peer-reviewed journal. The 40 pages are lengthy to many readers. The authors may consider rewriting some sections concisely or moving some sections to the appendix section, such as the discussion section. 

Another problem is the punctuation, which was missing in many places in the manuscript. It would cause some confusion to readers. 

I also notice a few grammar or editorial errors. A thorough proofreading is necessary. 

Comments on the Quality of English Language

The English is fluent. The quality of writing is high. However, the punctuation should be improved. 

Author Response

Comments

The referred manuscript presents a modified EBB model to predict the dynamic response of girder bridges under the load of passing trains, which has been validated by an FEA model and field-measured data. The good correlation shows that the model could successfully help assess bridge damage. Using the proposed model, the effect of pass frequencies on the resonance of different bridges and trains is discussed. The paper is well-written and structured. The research work is thoroughly discussed.

Author’s Response

We would like to thank the reviewer for the constructive comments.

Comments

However, the manuscript needs to be shorter for publication in a peer-reviewed journal. The 40 pages are lengthy to many readers. The authors may consider rewriting some sections concisely or moving some sections to the appendix section, such as the discussion section.

Author’s Response

Parts of the paper and a number of figures/tables have been moved into the Appendix section. Some additional sections had to be incorporated into the paper in response to a number of Reviewer 2's comments.

Comments

Another problem is the punctuation, which was missing in many places in the manuscript. It would cause some confusion to readers.

Author’s Response

The paper has been proof-read again and any punctuation mistakes have been corrected.

Comments

I also notice a few grammar or editorial errors. A thorough proofreading is necessary.

Author’s Response

The paper has been proof-read in detail and grammatical and editorial errors have been corrected in the updated version of the manuscript.

Reviewer 2 Report

Comments and Suggestions for Authors

Dear Authors,

Thank you for your manuscript. The paper focuses on the dynamic response of railway bridges to train configurations and their frequencies, which can lead to resonant responses and amplification of bridge displacement, stresses, and acceleration. This can result in increased fatigue damage and compromised train safety. The study presents a frequency response analysis for single span plate girder railway bridges and provides a generalized model for assessing the effect of wagon pass frequencies on bridge dynamic amplification. The mathematical model used is based on the Euler-Bernoulli Beam theory, and the results can be used by railway operators and bridge asset owners to determine optimal train speeds for different bridge and train configurations. However, the paper cannot be published in the present form. In the following, some remarks are proposed that should be addressed to improve the quality of the paper. Therefore, a major revision is required before the paper can be accepted for publication.

Abstract

The abstract could provide some details about the methodology used in the study, such as the specific equations or algorithms employed to determine the dynamic response of the bridge to moving loads. This would help readers understand the technical aspects of the research. 

The abstract could also mention the limitations of the study, such as any assumptions made or simplifications used in the mathematical model. This would provide a more comprehensive understanding of the research and its potential implications.

Introduction

The introduction could provide more background information on the significance of dynamic amplification in railway bridges and its impact on bridge performance and safety. This would help readers understand the importance of the research topic.

It could also mention the existing literature or studies on dynamic amplification of railway bridges, highlighting any gaps or limitations in the current knowledge. This would provide a context for the current research and justify its relevance.

Additionally, it could outline the objectives or research questions of the study, giving readers a clear understanding of what the research aims to achieve.

Lastly, it could provide a brief overview of the methodology or approach used in the study, giving readers an idea of how the research was conducted.

Bridge moving load models

It could provide a comprehensive overview of the existing bridge moving load models, including both analytical and numerical approaches, to give readers a clear understanding of the current state of the field. 

It could discuss the advantages and limitations of each model, highlighting their applicability to different types of bridges and train configurations. This would help readers assess the suitability of different models for their specific needs. 

It could also compare and contrast the accuracy and computational efficiency of different models, providing insights into their performance and potential areas for improvement. 

Additionally, it could discuss any recent advancements or emerging trends in bridge moving load modelling, such as the incorporation of advanced numerical techniques or the consideration of dynamic train-bridge interaction effects.

Case study plate girder bridges

This part lacks a detailed description of the specific characteristics and parameters of the case study plate girder bridges used in the analysis. This information is crucial for understanding the applicability and relevance of the study's findings.

It could provide more information on the selection criteria for the case study bridges, such as the reason for choosing low-frequency, light mass, and medium damping bridges. This would help readers understand the representativeness of the chosen bridges and the generalizability of the study's conclusions.

Additionally, it could discuss the specific challenges or issues faced in measuring and collecting data from the case study bridges, as well as the instrumentation used for the measurements. This would provide insights into the reliability and accuracy of the data used in the analysis.

Furthermore, it could discuss any limitations or assumptions made in the analysis of the case study bridges, such as the simplifications in the modeling approach or the neglect of certain factors. This would help readers assess the validity and robustness of the study's results.

Dynamic amplification factor

This section does not provide a comprehensive analysis of the factors influencing the dynamic amplification factor, such as bridge characteristics, train configurations, and loading conditions.

This section could benefit from discussing the methodologies used to calculate the dynamic amplification factor and their limitations. This would help readers understand the accuracy and reliability of the results.

Additionally, it could explore the effects of different train frequencies on the dynamic amplification factor and how it varies with respect to the natural frequency of the bridge. This would provide insights into the resonance phenomena and potential risks associated with specific train configurations.

It could also discuss the practical implications of the dynamic amplification factor, such as its impact on bridge displacement, stresses, acceleration, passenger comfort, noise levels, and train safety. This would help bridge engineers and operators make informed decisions regarding train speeds and bridge maintenance.

Section 3 EBB Dynamic Model Validation

The section does not provide a sufficiently detailed methodology for how the dynamic model validation was conducted. It should include a step-by-step explanation of the validation process, specifying the types of data used, the statistical methods for comparison, and any software or tools employed in the analysis.

There is a noticeable absence of detailed descriptions of the data sets used for validation. Information such as the source of the data, its temporal resolution, and the geographical context should be elaborated. This detail is crucial for understanding the applicability and limitations of the model.

The section lacks a comparative analysis between the model's predictions and real-world observations or data from other established models. Including such a comparison would significantly enhance the credibility of the model validation.

The metrics used to validate the model are not clearly defined or explained. It is essential to specify which statistical measures (e.g., RMSE, MAE, correlation coefficients) were used to assess the model's performance and why they were chosen.

Every model has limitations, but this section does not adequately discuss the limitations of the EBB dynamic model. A thorough understanding of these limitations is crucial for the appropriate application of the model.

The validation section would benefit from a sensitivity analysis to understand how changes in model parameters affect the output. This analysis is vital for identifying the model's robustness and areas for improvement.

The section does not address the quantification of uncertainty in the model's predictions. Including uncertainty analysis would provide a more comprehensive view of the model's reliability and accuracy.

Finally, there is insufficient discussion on how the findings from the validation process impact the overall understanding of the model's applicability. This discussion should include implications for future research, potential improvements to the model, and practical applications.

Section 4 Results and Discussion for Case Study Bridges

The discussion appears to be somewhat superficial and could be expanded to more thoroughly explore the significance of the findings. This includes a deeper analysis of how the results align or contrast with previous studies, and what these findings suggest about the broader field of bridge engineering and infrastructure resilience.

While the results are presented, there is a lack of detailed justification for the methodologies used in the case studies. Clarifying why certain methods were chosen and how they are suitable for the research questions at hand would strengthen this section.

The section would benefit from a more detailed statistical analysis of the results. This includes the use of appropriate statistical tests to validate the findings and a discussion on the statistical significance of the results.

The discussion on limitations and assumptions made during the case study analysis is either missing or not sufficiently detailed. Providing a clear overview of these limitations and assumptions would help readers understand the context and constraints of the study, enhancing its credibility.

The section could be improved by including a discussion on the broader implications of the findings for the field of bridge engineering, including potential recommendations for practice, policy, and future research. This would make the research more applicable and valuable to the reader.

There is a lack of explanation regarding the rationale behind the selection of case study bridges. Detailing why these particular bridges were chosen, including their relevance and representativeness, would provide more context for the reader.

Conclusion

The conclusions section lacks a clear, direct linkage to the initial objectives and research questions outlined at the beginning of the paper. Each conclusion should directly correspond to an objective or question, indicating whether it was achieved or answered, and how.

The conclusions are presented in a manner that is too general, lacking in specific details about the key findings. It is important for the conclusions to succinctly summarize the most significant results of the study, including any quantitative outcomes or qualitative insights that are critical to the study's contribution to the field.

While the section attempts to conclude the study, it does not adequately discuss the implications of the findings. This includes implications for future research, practice, policy, or theory within the field. Expanding on how the findings contribute to existing knowledge or open new avenues for investigation would add depth to the conclusions.

There is an absence of recommendations based on the study's findings. Providing specific, actionable recommendations for researchers would greatly enhance the utility and impact of the study.

The conclusions do not reflect on the limitations of the study. Acknowledging the limitations and the potential impact they might have on the interpretation of the findings is crucial for a balanced and credible conclusion.

The section lacks a clear outline of suggested directions for future research. Identifying specific gaps that the current study has uncovered, along with recommendations for how future studies could address these gaps, would be valuable.

The conclusions could be strengthened by including a reflective analysis of what the study achieved beyond the immediate findings, such as contributing to methodologies, theories, or conceptual frameworks within the field.

General comments

Across the paper, especially in sections related to model validation and case studies, there is a need for a more detailed explanation of methodologies used. This includes statistical methods, data collection procedures, and analytical tools.

A broader and more in-depth literature review could help in better contextualizing the study's findings within the existing body of knowledge. This includes discussing how the current study fills gaps or addresses unanswered questions from previous research.

Providing specific directions for future research based on the study's findings and limitations can guide subsequent work in the field and demonstrate the study's forward-looking relevance.

Comments on the Quality of English Language

Minor editing of English language required

Author Response

Abstract

Comments

The abstract could provide some details about the methodology used in the study, such as the specific equations or algorithms employed to determine the dynamic response of the bridge to moving loads. This would help readers understand the technical aspects of the research.

The abstract could also mention the limitations of the study, such as any assumptions made, or simplifications used in the mathematical model. This would provide a more comprehensive understanding of the research and its potential implications.

Author’s Response

The abstract has been re-written to include the methodology used, which is based on the Euler Bernoulli beam theory for a series of moving loads . We have stated that the method of solution was obtained using the inverse Laplace Carson transform, which allows a closed form solution to be obtained for the problem investigated.

In terms of the limitations, we have also stated that the model does not consider the mass of the train or its suspension system, but we have also provided a justification on why the approach adopted is considered to be acceptable. This is based on the practicality and efficiency that is needed when performing an initial assessment.

Introduction

Comments

The introduction could provide more background information on the significance of dynamic amplification in railway bridges and its impact on bridge performance and safety. This would help readers understand the importance of the research topic.

Author’s Response

A new section, i.e. sub-section, 1.1, has been included in the updated version of the paper which gives a brief overview of dynamic amplification and how this has evolved from the very early works in the railway industry. We have provided a basic insight on the needs to be addressed and how the DAF is dealt with in the current design codes, albeit very simply. 

Comments

It could also mention the existing literature or studies on dynamic amplification of railway bridges, highlighting any gaps or limitations in the current knowledge. This would provide a context for the current research and justify its relevance.

Author’s Response

The introduction has been split to provide additional subsections 1.2 and 1.3. Section 1.2 deals with how bridge dynamic response is modelled, providing a basic timeline of how this work has evolved and highlighting some of principal contributors in the field.  Section 1.3 specifically deals with studies related to dynamic amplification. We have concluded this section by highlighting  the reasons why the study of DAF is important and needs to be re-considered for current railway traffic, particularly in relation to fatigue where damage assessment can be under or overestimated when using DAF based on design codes.

In the next sub-section we have discussed and presented some of the more complex Train Bridge Interaction Models (TBI) and their advantages and challenges in terms of usability and applicability. 

Comments

Additionally, it could outline the objectives or research questions of the study, giving readers a clear understanding of what the research aims to achieve.

Lastly, it could provide a brief overview of the methodology or approach used in the study, giving readers an idea of how the research was conducted.

Author’s Response

To address the reviewer's comments, sub-section 1.5 has been included, which specifically highlights the need for further work in this field and how our paper's work contributes to fill this gap and concluding with a brief description of the methodology and the benefits of the model developed.

Bridge moving load models

Comments

It could provide a comprehensive overview of the existing bridge moving load models, including both analytical and numerical approaches, to give readers a clear understanding of the current state of the field.

It could discuss the advantages and limitations of each model, highlighting their applicability to different types of bridges and train configurations. This would help readers assess the suitability of different models for their specific needs.

It could also compare and contrast the accuracy and computational efficiency of different models, providing insights into their performance and potential areas for improvement.

Additionally, it could discuss any recent advancements or emerging trends in bridge moving load modelling, such as the incorporation of advanced numerical techniques or the consideration of dynamic train-bridge interaction effects.

Author’s Response

These review comments have been addressed within the updated introduction part of the paper by including sub-sections 1.2, 1.3 and 1.4, which provide a timeline of how bridge moving load models have evolved. At the end of each section we have summarised the key findings.

Case Study Plate Girder Bridges

Comments

This part lacks a detailed description of the specific characteristics and parameters of the case study plate girder bridges used in the analysis. This information is crucial for understanding the applicability and relevance of the study's findings.

It could provide more information on the selection criteria for the case study bridges, such as the reason for choosing low-frequency, light mass, and medium damping bridges. This would help readers understand the representativeness of the chosen bridges and the generalizability of the study's conclusions.

Additionally, it could discuss the specific challenges or issues faced in measuring and collecting data from the case study bridges, as well as the instrumentation used for the measurements. This would provide insights into the reliability and accuracy of the data used in the analysis.

Furthermore, it could discuss any limitations or assumptions made in the analysis of the case study bridges, such as the simplifications in the modelling approach or the neglect of certain factors. This would help readers assess the validity and robustness of the study's results.

Author’s Response

The case study bridges have been selected from Ref. [60] and the selection reflects bridges that are  typically found on the UK rail network. The selection of the bridges was made considering their span and structural parameters, that could potentially result in dynamic problems. We have also added a statement to highlight why we selected the bridge parameters for this study based on the proposed model and also highlighted how the model can be used to obtain other key information such as a global acceleration, whose limits are defined in the design codes.

Dynamic Amplification Factors

Comments

This section does not provide a comprehensive analysis of the factors influencing the dynamic amplification factor, such as bridge characteristics, train configurations, and loading conditions.

Author’s Response

The section has been amended to reflect the reviewer's comments.

Comments

This section could benefit from discussing the methodologies used to calculate the dynamic amplification factor and their limitations. This would help readers understand the accuracy and reliability of the results.

Author’s Response

The methodology and equations based on UK railway bridge assessment code are provided.

Comments

Additionally, it could explore the effects of different train frequencies on the dynamic amplification factor and how it varies with respect to the natural frequency of the bridge. This would provide insights into the resonance phenomena and potential risks associated with specific train configurations.

Author’s Response

Additional statements to address the comments raised have been provided. We have re-iterated how wagon pass frequencies and its multiples can affect the bridge dynamic response.

Comments

It could also discuss the practical implications of the dynamic amplification factor, such as its impact on bridge displacement, stresses, acceleration, passenger comfort, noise levels, and train safety. This would help bridge engineers and operators make informed decisions regarding train speeds and bridge maintenance.

Author’s Response

We have provided statements on how fatigue, deck accelerations and noise would be affected by dynamic amplification and stated how this work can provide a means by which engineers can make informed decisions.

EBB Dynamic Model Validation

Comments

The section does not provide a sufficiently detailed methodology for how the dynamic model validation was conducted. It should include a step-by-step explanation of the validation process, specifying the types of data used, the statistical methods for comparison, and any software or tools employed in the analysis.

There is a noticeable absence of detailed descriptions of the data sets used for validation. Information such as the source of the data, its temporal resolution, and the geographical context should be elaborated. This detail is crucial for understanding the applicability and limitations of the model.

Author’s Response

To address this comment further additional detail on verification of the model based on available resources and data along with description of these information is provided in the revised manuscript.

Comments

The section lacks a comparative analysis between the model's predictions and real-world observations or data from other established models. Including such a comparison would significantly enhance the credibility of the model validation.

Author’s Response

A comparison between our model's predictions and real world observation data has been carried out by using independently obtained response data. Comparison with other model is not feasible and aligned with the scope of the study here as they take into account other parameters of the bridge response, not similar to our model.

Comments

The metrics used to validate the model are not clearly defined or explained. It is essential to specify which statistical measures (e.g., RMSE, MAE, correlation coefficients) were used to assess the model's performance and why they were chosen.

Author’s Response

In view of the nature of this study and its objectives, statistical analysis has not been conducted as is not aligned with the purpose of the investigation and considered outside the scope of this study.

Comments

Every model has limitations, but this section does not adequately discuss the limitations of the EBB dynamic model. A thorough understanding of these limitations is crucial for the appropriate application of the model.

Author’s Response

We have added additional statements to further highlight the limitations of the model, but also how the model is of value in providing vital and actionable information to enable engineers to make informed decisions relatively easily and quickly.

Comments

The validation section would benefit from a sensitivity analysis to understand how changes in model parameters affect the output. This analysis is vital for identifying the model's robustness and areas for improvement.

The section does not address the quantification of uncertainty in the model's predictions. Including uncertainty analysis would provide a more comprehensive view of the model's reliability and accuracy.

Author’s Response

The authors acknowledge the value of conducting a sensitivity analysis on model validation parameters, as it offers insights into the influence of each parameter. However, in this particular study, the model primarily functions as a demonstrator of an alternative approach. Consequently, while validating the model remains crucial, performing a sensitivity analysis is deemed outside the study's scope. Given the study's length, integrating a sensitivity analysis would not be feasible.

Comments

Finally, there is insufficient discussion on how the findings from the validation process impact the overall understanding of the model's applicability. This discussion should include implications for future research, potential improvements to the model, and practical applications.

Author’s Response

In the summary section, we have added a general discussion on the limitations of the  model, identifying other key parameters which can be considered. However, whether these have any impact on fatigue damage, which is the focus of this paper, will require further work as there is little information on this area in the literature.

Results and Discussion for Case Study Bridges

Comments

The discussion appears to be somewhat superficial and could be expanded to more thoroughly explore the significance of the findings. This includes a deeper analysis of how the results align or contrast with previous studies, and what these findings suggest about the broader field of bridge engineering and infrastructure resilience.

Author’s Response

The results discussion section is now revised to further highlight the significance of findings with reflection on its implication for current practice in contrast to alternative complex numerical models and other previous studies in this field.

Comments

While the results are presented, there is a lack of detailed justification for the methodologies used in the case studies. Clarifying why certain methods were chosen and how they are suitable for the research questions at hand would strengthen this section.

Author’s Response

We have revised the methodology section to provide a clearer explanation of why the specific methods were chosen and elaborating on the rationale behind our proposed approach (i.e., efficiency, ease of explainability and versatility of the approach).

Comments

The section would benefit from a more detailed statistical analysis of the results. This includes the use of appropriate statistical tests to validate the findings and a discussion on the statistical significance of the results.

Author’s Response

We believe that statistical analysis is outside the scope of this paper and not aligned with its objectives.

Comments

The discussion on limitations and assumptions made during the case study analysis is either missing or not sufficiently detailed. Providing a clear overview of these limitations and assumptions would help readers understand the context and constraints of the study, enhancing its credibility.

The section could be improved by including a discussion on the broader implications of the findings for the field of bridge engineering, including potential recommendations for practice, policy, and future research. This would make the research more applicable and valuable to the reader.

Author’s Response

We have added additional statements to further highlight the limitations of the model, but also how the model is of value in providing vital and actionable information to enable engineers to make informed decisions relatively easily and quickly.

Comments

There is a lack of explanation regarding the rationale behind the selection of case study bridges. Detailing why these particular bridges were chosen, including their relevance and representativeness, would provide more context for the reader.

Author’s Response

The rationale for selecting the bridges has been explained in Section 2.2, and we have made some additional amendments to reflect the reviewer's comment.

Conclusion

Comments

The conclusions section lacks a clear, direct linkage to the initial objectives and research questions outlined at the beginning of the paper. Each conclusion should directly correspond to an objective or question, indicating whether it was achieved or answered, and how.

Author’s Response

We have amended the conclusion by highlighting what the aims and objectives of this study are and how they have been addressed in the course of this study.

Comments

The conclusions are presented in a manner that is too general, lacking in specific details about the key findings. It is important for the conclusions to succinctly summarize the most significant results of the study, including any quantitative outcomes or qualitative insights that are critical to the study's contribution to the field

Author’s Response

The conclusion section is revised to further highlight the findings on which trains are more likely to cause an amplification of the dynamic response, and how these also affect the multiples of the wagon pass frequency.

Comments

While the section attempts to conclude the study, it does not adequately discuss the implications of the findings. This includes implications for future research, practice, policy, or theory within the field. Expanding on how the findings contribute to existing knowledge or open new avenues for investigation would add depth to the conclusions.

There is an absence of recommendations based on the study's findings. Providing specific, actionable recommendations for researchers would greatly enhance the utility and impact of the study.

The conclusions do not reflect on the limitations of the study. Acknowledging the limitations and the potential impact they might have on the interpretation of the findings is crucial for a balanced and credible conclusion.

The section lacks a clear outline of suggested directions for future research. Identifying specific gaps that the current study has uncovered, along with recommendations for how future studies could address these gaps, would be valuable.

Author’s Response

We have added a sub section to highlight the limitations of the model and recommendations for further work.

Comments

The conclusions could be strengthened by including a reflective analysis of what the study achieved beyond the immediate findings, such as contributing to methodologies, theories, or conceptual frameworks within the field.

Author’s Response

The introductory paragraph has been amended to reflect this comment.

General Comments

Comments

Across the paper, especially in sections related to model validation and case studies, there is a need for a more detailed explanation of methodologies used. This includes statistical methods, data collection procedures, and analytical tools.

Author’s Response

Further explanations have been added to strengthening the existing ones based on the reviewer's comment. Statistical analysis, however, as addressed earlier, is outside the scope of this work.

Comments

A broader and more in-depth literature review could help in better contextualizing the study's findings within the existing body of knowledge. This includes discussing how the current study fills gaps or addresses unanswered questions from previous research.

Author’s Response

A broader literature survey has been included in the updated version of the paper and the Introduction has been split in distinct sub-sections to address the reviewer's comments.

Comments

Providing specific directions for future research based on the study's findings and limitations can guide subsequent work in the field and demonstrate the study's forward-looking relevance.

Author’s Response

A sub-section has been added to highlight the limitations of the model and sc

Reviewer 3 Report

Comments and Suggestions for Authors

The work presents a topic that is of interest for the safety of railway traffic.

The title of the article correctly summarizes the content of the research.

Under the aspect of the research carried out, the level of approach is noteworthy, both analytically and through FE modeling. An observation could be the large number of case studies, which considerably increases the volume of the work, but, considering that the same aspects are analyzed in all of them, it can be appreciated that the work is exhaustive and the results ensure adequate substantiation of the stated purpose.

The conclusions are consistent with the results and clearly show that the goal of the research has been achieved.

The novelty of the article is given by the approach to the research and the conclusions formulated based on the results obtained.

A note to the authors is also necessary related to the English translation, where in certain situations, it contains repeated words. These repeated words should be considered and replaced where possible, even if the terms, (which are repeated), refer to parameters with established names.

Comments on the Quality of English Language

Minor editing of English language required.

Author Response

Comments

Under the aspect of the research carried out, the level of approach is noteworthy, both analytically and through FE modelling. An observation could be the large number of case studies, which considerably increases the volume of the work, but, considering that the same aspects are analyzed in all of them, it can be appreciated that the work is exhaustive and the results ensure adequate substantiation of the stated purpose.

Author’s Response

We have included a number of case studies in order to investigate the range of the applicability of the developed model. We agree that this makes the work exhaustive and complete, increasing confidence of the obtained predictions and understand the limitations. 

Comments

A note to the authors is also necessary related to the English translation, where in certain situations, it contains repeated words. These repeated words should be considered and replaced where possible, even if the terms, (which are repeated), refer to parameters with established names.

Author’s Response

The paper has been proof-read in detail and the English language has been polished.

Round 2

Reviewer 2 Report

Comments and Suggestions for Authors

Accept